# Tuning Mixed Input Hyperparameters on the Fly for Efficient Population Based AutoRL

**Jack Parker-Holder**[*]
University of Oxford

**Vu Nguyen**
Amazon, Australia

**Shaan Desai**
University of Oxford

**Stephen Roberts**
University of Oxford

## Abstract

Despite a series of recent successes in reinforcement learning (RL), many RL algorithms remain sensitive to hyperparameters. As such, there has recently been interest in the field of AutoRL, which seeks to automate design decisions to create more general algorithms. Recent work suggests that population based approaches may be effective AutoRL algorithms, by learning hyperparameter schedules on the fly. In particular, the PB2 algorithm is able to achieve strong performance in RL tasks by formulating online hyperparameter optimization as time varying GP-bandit problem, while also providing theoretical guarantees. However, PB2 is only designed to work for *continuous* hyperparameters, which severely limits its utility in practice. In this paper we introduce a new (provably) efficient hierarchical approach for optimizing *both continuous and categorical* variables, using a new time-varying bandit algorithm specifically designed for the population based training regime. We evaluate our approach on the challenging Procgen benchmark, where we show that explicitly modelling dependence between data augmentation and other hyperparameters improves generalization.

## 1 Introduction

Reinforcement Learning (RL [75]) is a paradigm whereby agents learn to make sequential decisions through trial and error. In the past few years there have been a series of breakthroughs using RL in games [70, 44, 6] and robotics [50, 33], which have led to a surge of interest in the machine learning community. Beyond these examples, RL offers the potential to impact vast swathes of society, from autonomous vehicles to robotic applications in healthcare and industry.

Despite the promising results, RL is notoriously difficult to use in practice. In particular, RL algorithms are incredibly sensitive to hyperparameters [23, 2, 18, 27, 9, 47], with careful tuning often the difference between success and failure. Furthermore, as new and more complex algorithms are introduced, the search space continues to grow [2]. As a result, it is often challenging to reproduce RL results given the significant number of moving parts in modern algorithms [2]. This makes it almost impossible to apply these methods in novel settings, where optimal hyperparameters are unknown. On the other hand, the capability of current methods may be understated, as better configurations could boost performance [10, 84].

In this work we focus on the recent impressive results for *Population Based Training* (PBT, [29, 39]), which has demonstrated strong performance in a variety of prominent RL settings [65, 84, 43, 19, 28]. PBT works by training agents in parallel, with an evolutionary outer loop optimizing hyperparameters, periodically replacing weaker agents with perturbations of stronger ones. However, since PBT relies on random search to explore the hyperparameter space, it requires a large population size which can be prohibitively expensive for small and even medium-sized labs.

---

[*]Correspondence to jackph@robots.ox.ac.uk.

35th Conference on Neural Information Processing Systems (NeurIPS 2021).

Table 1: The components of related approaches, and their relative trade-offs.

| Algorithm | Population Based? | Efficient Continuous? | Efficient Categorical? |
|---|---|---|---|
| PBT/PBA [29, 25] | ✓ | ✗ | ✗ |
| PB2 [52] | ✓ | ✓ | ✗ |
| CoCaBO [63] | ✗ | ✓ | ✓ |
| This work | ✓ | ✓ | ✓ |

In order to achieve similar success with a smaller computational budget, the recent *Population Based Bandits* (PB2, [52]) algorithm improved sample efficiency by introducing a probabilistic exploration step, backed by theoretical guarantees. Unfortunately, a key limitation of PB2 is that it only addresses the problem of efficiently selecting *continuous* hyperparameters, inheriting the random search method for *categorical* variables from the original PBT. This is not only inefficient but also crucially ignores potential dependence between continuous and categorical variables. Since the Gaussian Process (GP, [58]) model is completely unaware of the changing categories, it is unable to differentiate between trials that may have completely different categorical hyperparameters. In this paper we introduce a new hierarchical approach to PB2, which can efficiently model *both* continuous and categorical variables, with theoretical guarantees. Our main contributions are the following:

**Technical**: We introduce a new PB2 explore step which can *efficiently* choose between *both* categorical and continuous hyperparameters in a population based training setup. In particular, we propose a new time-varying batch multi-armed bandit algorithm, and introduce two hierarchical algorithms which condition on the selected categorical variables. We show our new approach achieves sublinear regret, extending the results for the continuous case with PB2.

**Practical**: We scale our approach to test generalization on the Procgen benchmark [13], an active area of research. We demonstrate improved performance when explicitly modelling dependence between data augmentation type and continuous hyperparameters (such as learning rate) vs. baselines using random search to select the data augmentation.

## 2   Related Work

This work contributes to the emerging field of AutoRL [11], which seeks to automate elements of the reinforcement learning (RL) training procedure. Automating RL hyperparameter tuning has been studied since the 1980s [4], with a surge in recent interest due to the increasing complexity of modern algorithms [84, 54]. The scope for AutoRL is broad, from using RL in novel real-world problems where hyperparameters are unknown [64], to improving the performance of existing methods [10]. Recent successes in AutoRL include learning differentiable hyperparameters with meta-gradients [81, 83], while it has recently been shown it is even possible to learn algorithms [49, 12].

In this paper we focus on the class of *Population Based Training* (PBT [29]) methods. Inspired by how humans run experiments, PBT trains agents in parallel and periodically replaces the weakest agents with variations of the stronger ones, learning a hyperparameter *schedule* on the fly, in a single training run. While PBT is applicable in any AutoML [26] setting, it has been particularly impactful in deep RL [19, 84, 65]. PBT was recently shown to work well with a shared replay buffer [20], but this only applies for off policy methods. This paper builds on the recently introduced *Population Based Bandits* (PB2, [52]) algorithm, which employs a Bayesian Optimization (BO, [8, 24, 71, 17, 63, 46]) approach to select hyperparameters during the course of training. Concretely PB2 casts the hyperparameter selection step of PBT as a batch time-varying GP bandit optimization problem [7]. The strength of PB2 lies in finding optimal configurations with a small population size, yet at present it has only been developed for continuous hyperparameters, relying on random search for the categorical variables.

Categorical variables are prominent in RL, for example the choice of exploration strategy [77], or even algorithm class. A prominent recent categorical variable is the choice of data augmentation, which has been shown to significantly improve efficiency and generalization in RL [37, 35, 38, 80], reducing observational overfitting [73]. Recent work introduced *automatic data augmentation* [57], which shows significant improvement over a static baseline by learning the data augmentation on the fly (with a single agent). However, this approach relies on a grid search over new hyperparameters, keeping all others fixed. We take inspiration from this result and learn *both* data augmentation and baseline RL algorithm hyperparameters jointly on the fly, with a population of agents. PBT has also been extended to data augmentation (PBA, [25]) with strong results in supervised learning tasks. However, in this setting the data augmentation was set up as a continuous variable. A high level summary of differences vs. prior work is shown in Table 1.

## 3    The Case for AutoRL

In this section we introduce the reinforcement learning (RL, [75]) paradigm, before making the case for automating hyperparameter selection for policy gradient algorithms.

### 3.1    Reinforcement Learning Background

A Markov Decision Process (MDP) is a tuple $(\mathcal{S}, \mathcal{A}, P, R, \gamma)$, where for each time step $t = 0, 1, 2 \ldots$ the environment provides the agent with an observation $s_t \in \mathcal{S}$, the agent responds by selecting an action $a_t \in \mathcal{S}$, and then the environment provides the next reward $r_t$, discount $\gamma_{t+1}$, and state $s_{t+1}$. Reinforcement learning (RL) considers the problem of learning a policy $\pi$ which selects actions that maximize the expected total discounted reward [5, 76, 75]. While our method can be applied to **any** RL algorithm, for simplicity we focus our discussion on *policy gradient* algorithms, which directly seek to maximize the policy with respect to expected reward. For a policy $\pi : \mathcal{S} \rightarrow \mathcal{A}$ parameterized by $\theta$ the objective is as follows:

$$J(\pi_\theta) = \mathbb{E}_{\tau \sim \pi_\theta} R(\tau)$$

where $\tau = \{s_1, a_1, r_1, \ldots, s_H, a_H, r_H\}$ for some horizon $H$. Of course, we wish to maximize this, so optimize the policy by taking the following gradient steps:

$$\theta_{t+1} = \theta_t + \alpha \nabla_\theta J(\pi_{\theta_t}).$$

### 3.2    Case study: Proximal Policy Optimization

Proximal Policy Optimization (PPO [68]) is one of the most widely used RL algorithms, achieving strong performance in continuous control problems, and even scaling to large scale games [6]. Building on Trust Region Policy Optimization (TRPO [66]), the success of PPO comes from using a clipped loss function as follows:

$$\mathcal{L}_{\text{PPO}}(\theta) = \min\left( \frac{\pi_\theta(a|s)}{\pi_\mu(a|s)} A^{\pi_\mu}, g(\theta, \mu) A^{\pi_\mu} \right), \text{ where } g(\theta, \mu) = \text{clip}\left( \frac{\pi_\theta(a|s)}{\pi_\mu(a|s)}, 1 - \epsilon, 1 + \epsilon \right)$$

for a previous policy $\pi_\mu$, an advantage function $A$ and a clipping hyperparameter $\epsilon$. As can be seen, PPO's success hinges on several new hyperparameters. As well as the learning rate $\alpha$, clip parameter $\epsilon$, decay $\gamma$, there is also often another hyperparameter in the advantage function $\lambda$ [67]. As if this wasn't enough, a recent study showed we also need to consider *code level* hyperparameters [18] after recent work found nine implementation details (such as reward scaling) had a significant impact on performance. Thus, it is clear that while PPO can be effective, it requires careful tuning if it is to be used effectively for novel problems.

In addition, a recent study investigating actor-critic algorithms [2] found that not only are there many more hyperparameters than previously thought (for example weight initialization strategy), but their optimal values are dependent on one another, producing a combinatorially large search space. This problem is only increasing as novel methods extending upon PPO they almost always add *additional* hyperparameters [57]. Thus, if it wasn't enough that PPO itself has tens of hyperparameters, new additions may render all previous values sub-optimal since they may be dependent on the new variables. The main hypothesis in this paper is that we can improve performance of baseline algorithms by *jointly* learning several hyperparameters *on the fly*.

## 4    Population Based Bandits

We consider the problem of optimizing a population of agents in parallel, dynamically adapting their weights and hyperparameters such that strong performance can be achieved in a single training run. Following existing work [29, 52], we consider two sub-routines, explore and exploit. We train for a total of $T$ steps, evaluating performance every $t_{\text{ready}} < T$ steps. For the exploit step, the weights of the bottom agents are replaced by those from a randomly sampled agent from the set of best performing agents, in what is called *truncation selection*. The next step is to select new hyperparameters, with the explore step. The full procedure is shown in Algorithm 1 and Fig. 1.

**Algorithm 1** Population Based Training (Bandits)

---

**Initialize:** Population network weights, hyperparameters, empty dataset.
**for (in parallel)** $t = 1, \ldots, T - 1$ **do**
1. **Train Models**
2. **Evaluate Models and Record Data**
3. If $t \mod t_{\text{ready}} = 0$:

- **Exploit:** Replace the weights of weaker agents with those from stronger agents.
- **Explore:** If weights were replaced, select new hyperparameters.

**Return the best trained model** $\theta$

---

The key difference between Population Based Training (PBT, [29]) and Population Based Bandits (PB2, [52]) is the explore step. PBT selects new hyperameters using random perturbations, hence it often performs poorly in a resource constrained setting. By contrast PB2 (blue in Alg. 1) models the data from previous trials to efficiently explore the hyperparameter space. We focus on the PB2 framework and introduce new algorithms for mixed-input hyperparameters.

## 4.1 Online Hyperparameter Selection as GP-Bandit Optimization

We consider the problem of selecting optimal hyperparameters, $x_t^b$, from a compact, convex subset $\mathcal{D} \in \mathbb{R}^d$ where $d$ is the number of hyperparameters. Here the index $b$ refers to the $b$th agent in a population, and the subscript $t$ represents the number of timesteps elapsed during the training of a neural network. In particular, we consider the *schedule* of optimal hyperparameters over time $\left(x_t^b\right)_{t=1,\ldots T}$. Let $F_t(x_t)$ be an objective function under a given set of hyperparameters at timestep $t$. Here we take $F_t(x_t)$ to be the reward for a deep RL agent. Our goal is to maximize the final performance $F_T(x_T)$.

Following [52] we formulate this problem as optimizing the time-varying black-box reward function $f_t$, over $\mathcal{D}$. Every $t_{\text{ready}}$ steps, we observe and record noisy observations, $y_t = f_t(x_t) + \epsilon_t$, where $\epsilon_t \sim \mathcal{N}(0, \sigma^2 \mathbf{I})$ for some fixed $\sigma^2$. Note the function $f_t$ represents the

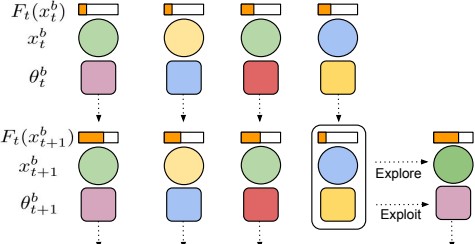

Figure 1: An overview of the PBT/PB2 framework. We have a population of $B$ agents, each with weights $\theta_t^b$ and hyperparameters $x_t^b$. $t$ refers to the stage of training, with each increment corresponding to training for $t_{\text{ready}}$ steps. The performance of the agent is represented by $F_t$. At each $t$, the weights of the worst performing agents are replaced (exploit), where weights are copied from better agents. The new hyperparameters are then selected using an explore procedure, the focus of this work.

*change* in $F_t$ after training for $t_{\text{ready}}$ steps, i.e. $F_t - F_{t-t_{\text{ready}}}$. We define the best choice at each timestep as $x_t^* = \arg\max_{x_t \in \mathcal{D}} f_t(x_t)$, and so the **regret** of each decision as $r_t = f_t(x_t^*) - f_t(x_t)$. Minimizing the regret of each decision is equivalent to maximizing the final reward [52], i.e. $\max F_T(x_T) = \min \sum_{t=1}^{T} r_t(x_t)$. We now discuss how PB2 seeks to minimize regret for *continuous* hyperparameters.

## 4.2 Parallel Gaussian Process Bandits for a Time-Varying Function

PB2 models the data using a Gaussian Process (GP, [58]), thus assumes the data follows a Gaussian distribution with mean $\mu_t(x')$ and variance $\sigma_t^2(x')$ as:

$$\mu_t(x') := \mathbf{k}_t(x')^T (\mathbf{K}_t + \sigma^2 \mathbf{I})^{-1} \mathbf{y}_t \tag{1}$$

$$\sigma_t^2(x') := k(x', x') - \mathbf{k}_t(x')^T (\mathbf{K}_t + \sigma^2 \mathbf{I})^{-1} \mathbf{k}_t(x'), \tag{2}$$

where $\mathbf{K}_t := \{k(x_i, x_j)\}_{i,j=1}^t$ and $\mathbf{k}_t := \{k(x_i, x_t')\}_{i=1}^t$. The GP predictive mean and variance above will later be used to represent the exploration-exploitation trade-off in making decisions in the presence of uncertainty. The key insight in PB2 is that it is possible to consider the time-varying nature of neural network hyperparameters. This is consistent with empirical results, for example, the use of learning rate schedules in almost all large scale neural network models [36]. PB2 follows [7] by modeling the reward function under the time-varying setting as follows:

$$f_1(x) = g_1(x), \quad f_{t+1}(x) = \sqrt{1-\omega} f_t(x) + \sqrt{\omega} g_{t+1}(x) \quad \forall t \geq 2,$$

where $g_1, g_2, ...$ are independent random functions with $g \sim GP(0, k)$ and $\omega \in [0, 1]$ models how the function varies with time, such that if $\omega = 0$ we return to GP-UCB and if $\omega = 1$ then each evaluation is completely independent. This leads to the extensions of Eqs. (1) and (2) using the new covariance matrix $\tilde{\mathbf{K}}_t = \mathbf{K}_t \circ \mathbf{K}_t^{\text{time}}$ where $\mathbf{K}_t^{\text{time}} = [(1 - \omega)^{|i-j|/2}]_{i,j=1}^T$ and $\tilde{\mathbf{k}}_t(x) = \mathbf{k}_t \circ \mathbf{k}_t^{time}$ with $\mathbf{k}_t^{time} = [(1 - \omega)^{(T+1-i)/2}]_{i=1}^T$. Here $\circ$ refers to the Hadamard product.

**Selecting hyperparameters for parallel agents.** A key observation in [17] is that since a GP's variance (Eqn. 2) does not depend on $y_t$, the acquisition function can account for incomplete trials by updating the uncertainty at the pending evaluation points. Recall $x_t^b$ is the $b$-th point selected in a batch, after $t$ timesteps. This point may draw on information from $t + (b - 1)$ previously selected points. In the single agent, sequential case, we set $B = 1$ and recover $t, b = t - 1$. Thus, at the iteration $t$, we find a next batch of $B$ samples $[x_t^1, x_t^2, ...x_t^B]$ by sequentially maximizing the following acquisition function:

$$x_t^b = \arg\max_{x \in \mathcal{D}} \mu_{t,1}(x) + \sqrt{\beta_t}\sigma_{t,b}(x), \forall b = 1, ...B \tag{3}$$

for $\beta_t > 0$. In Eqn. (3) we have the mean from the previous batch ($\mu_{t,1}(x)$) which is fixed, but can update the uncertainty using our knowledge of the agents currently training ($\sigma_{t,b}(x)$). This significantly reduces redundancy, as the model is able to explore distinct regions of the space.

To summarize, the PB2 explore step works by optimizing Eqn. (3) to select a batch of new parameters. However, it is clear to see that this approach is not equipped with a means to select *categorical* variables. In fact, the existing PB2 algorithm inherits the random selection of categories from PBT. To reflect this, from this point onwards we refer to PB2 from [52] as **PB2-Rand**.

# 5  Efficient Selection of Continuous and Categorical Variables

We propose a new class of hierarchical population based approaches for handling continuous and categorical variables. To the best of our knowledge, this is the first provably efficient approach for optimizing the categorical variables in the PBT family [29, 52]. First we introduce a new time-varying version of the parallel EXP3 algorithm, which we call TV.EXP3.M, used in all of our methods to select categories. We then present three alternative approaches to subsequently select the continuous variables in a hierarchical fashion, varying in their degree of dependence on the categorical choices.

## 5.1  Time-varying Parallel EXP3

We introduce TV.EXP3.M, a new algorithm for parallel multi-armed bandits (MAB) in a time-varying setting with adversarial feedback. TV.EXP3.M is an extension of the multiple play EXP3 algorithm (or EXP3.M) [78] for time-varying rewards. At each round, TV.EXP3.M takes multiple actions from the set of all possible choices [78], making it possible to use this approach with a population of agents. See Algorithm 2 for a high level summary and Algorithm 5 in Sec. D.1 for further details.

---

**Algorithm 2** TV.EXP3.M (brief)

---

Input: $\gamma = \sqrt{\frac{C \ln(C/B)}{(e-1)BT}}, \alpha = \frac{1}{T}, C$ #categorical choice, $T$ #max iteration, $B$ #multiple play

1: Init $w_c = 1, \forall c = 1...C$ and denote $\eta = (\frac{1}{B} - \frac{\gamma}{C})\frac{1}{1-\gamma}$
2: **for** $t = 1$ to $T$ **do**
3:     Normalize $w$ to prevent from over exploitation
4:     Compute the prob for each arm $p_t^c, \forall c$
5:     Select a batch of $B$ choices $S_t = [c_t^1, c_t^2, ..., c_t^B] = \text{DepRound}\left(B, [p_t^1 p_t^2 ....p_t^C]\right)$
6:     Observe the reward after evaluating $S_t$
7:     Update the weights using $\eta$ for each arm $w_t^c, \forall c$
8: **end for**

---

We treat each categorical choice as an arm in the bandit setting. Given a collection of $C$ categories, each includes a reward probability, we select a batch of $B$ points

$$S_t = [c_t^1, c_t^2, ..., c_t^B] = \text{TV.EXP3.M}(D_{t-1}). \tag{4}$$

To efficiently select a set of $B$ distinct arms from $[C]$, we use the technique of dependent rounding (DepRound) while satisfying the condition that each arm $c$ is selected with probability $p_c$ exactly [21]. TV.EXP3.M is provably efficient, with a sublinear regret bound.

**Theorem 1.** *Set $\alpha = \frac{1}{T}$ and $\gamma = \min\left\{1, \sqrt{\frac{C\ln(C/B)}{(e-1)BT}}\right\}$, we assume the reward distributions changes at arbitrary instances, but the total number of change points is no more than $V \ll \sqrt{T}$ times. The expected regret of TV.EXP3.M satisfies the following sublinear bound*

$$\mathbb{E}\left[R_{TB}\right] \leq [1 + e + V]\sqrt{(e-1)\frac{CT}{B}\ln\frac{CT}{B}}.$$

## 5.2 New Exploration Strategies

Now we are ready to present two novel hierarchical exploration strategies. Both methods use TV.EXP3.M to select categories before subsequently choosing continuous variables.

| **Algorithm 3** PB2-Mult: explore |
| --- |
| 1. Select $\{h_t^b\}_{b=1}^B$ using TV.EXP3.M |
| 2. Filter dataset. |
| 3. Select $x_t^b$ by optimizing Eqn. (3). |
| **Output:** $\{h_t^b\}_{b=1}^B, \{x_t^b\}_{b=1}^B$ |

| **Algorithm 4** PB2-Mix: explore |
| --- |
| 1. Select $\{h_t^b\}_{b=1}^B$ using TV.EXP3.M |
| 2. Select $x_t^b$ by optimizing Eqn. (3), using the kernel from Eqn. (5) |
| **Output:** $\{h_t^b\}_{b=1}^B, \{x_t^b\}_{b=1}^B$ |

**PB2-Mult (Algorithm 3):** We consider a special version of dependency in which the presence of the continuous variables entirely depends on if the corresponding category is switched on. This not only allows us to fully incorporate dependence between hyperparameters, but also allows us to have different hyperparameters depending on the category. For example, if our categorical variable is the choice of neural network optimizer, then the current optimal learning rate may vary significantly if using Adam [34] or SGD. Furthermore, Adam requires additional hyperparameters, $\beta_1$ and $\beta_2$. We handle this dependency by constructing *multiple* time-varying GP surrogate models, each corresponding to a choice of category. We call this method **PB2-Mult**. PB2-Mult is the best choice if the continuous variables are highly dependent on the category, or if there are differing numbers of continuous variables for each category.

**PB2-Mix (Algorithm 4):** We also propose a mechanism to incorporate dependence between continuous and categorical variables *directly into the GP kernel*, in what we call **PB2-Mix**. To do this, we extend the joint continuous-categorical kernel in CoCaBO [63] to the time-varying setting. Denote $z = [x, c]$, we have:

$$k_z(z, z') = (1 - \lambda)\left(\tilde{\mathbf{k}}_t(x) + k_{ct}\right) + \lambda\tilde{\mathbf{k}}_t(x)k_{ct} \tag{5}$$

where $k_{ct} = \frac{\sigma_2}{C}\sum\mathbb{I}(c, c') \times \mathbf{k}_t^{time}$ and $\tilde{\mathbf{k}}_t(x), \mathbf{k}_t^{time}$ are defined in Sec. 4.2. In the above formulation, we first select a batch of $B$ categorical choices $c_t^b, \forall b \leq B$ using TV.EXP3.M. Then, we utilize the kernel in Eqn. (5) to learn the time-varying GP model to select the next batch of continuous points $x_t^b, \forall b \leq B$ following Eqn. (3), as in PB2-Rand. Each agent $b$ will then train with $[x_t^b, c_t^b]$. We do not need to set the parameter $\lambda$ as it can be optimized alongside the other kernel parameters (see the Appendix, Sec: D.3). Thus, PB2-Mix in principle achieves the best of both worlds: it is able to learn the relative importance of the continuous and categorical kernels, while including all of the data.

## 5.3 Main theoretical results

We extend the sublinear regret bound from PB2-Rand to PB2-Mult, providing performance guarantees. Under the categorical-continuous setting, we show the equivalence that minimizing the cumulative regret $R_T$ is equivalent to maximizing the reward, presented in Lem. 4 in the appendix. Our main theoretical result is to derive the regret bound for PB2-Mult.

**Theorem 2.** *The PB2-Mult algorithm has cumulative regret bounded with high probability*

$$\mathbb{E}\left[R_{TB}\right] \lesssim \mathcal{O}\left(V\sqrt{\frac{CT}{B}\ln T} + \sqrt{\frac{T_{c_t^*}^2}{\tilde{N}B}(\gamma_{\tilde{N}B} + [\tilde{N}B]^3\omega)}\right)$$

*for $\tilde{N} \to T_{c_t^*}$, $\omega \to 0$, and $V \ll \sqrt{T}$.*

From Theorem 1, we have $T_{c_t^*} \to \infty$ when $T \to \infty$. Asymptotically PB2-Mult achieves sublinear convergence rate, i.e., $\lim_{T\to\infty}\frac{\mathbb{E}[R_{TB}]}{T} = 0$. We refer to Sec. D.2 in the Appendix for proofs.

## 6 Experiments

The primary goal of our experiments is to test the efficacy of PB2-Mult and PB2-Mix against PB2-Rand, the primary baseline. We consider two settings: first, a synthetic task where we know that the continuous and categorical variables exhibit strong dependence. We then move to the AutoRL setting, where we apply them to select both continuous hyperparameters and categorical data augmentation type for challenging pixel-based procedurally generated environments [57].

### 6.1 Synthetic Function with Dependency

We begin with a simple synthetic function with one continuous and one categorical variable. The categorical variable has two options $h_t^b \sim \{\sin, \cos\}$ and the continuous variable has bounds $x_t^b \sim [0, \frac{\pi}{2}]$. The blackbox function $f : [x, h] \to \mathbb{R}$ returns $h(x)$, and the regret is equal to $1 - f_t(.)$. It is clear to see the function is maximized (and regret minimized) at $\{\sin, \frac{\pi}{2}\}$ and $\{\cos, 0\}$. We compare PB2-Mult and PB2-Mix against PB2-Rand, PBT and Random Search. At each iteration, the worst performing agent(s) is replaced by the best and the categories and continuous variables are re-selected. We run the example 20 times, with a population size of 4, 8 or 12 agents.

Fig 2 shows the mean cumulative regret, with the standard error (sem) shaded. As we see, random search has linear regret, while PBT and PB2-R both provide incremental improvements. Our new approaches outperform, led by PB2-Mult. Thus, we see that when there is strong dependence between the two hyperparameters, the methods which explicitly model

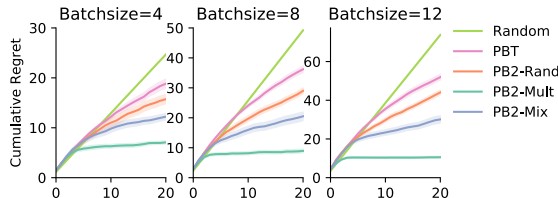

Figure 2: Mean $\pm 1$sem cumulative regret for 20 runs.

this significantly outperform. For details, see the following notebook http://bit.ly/synthetic_func, including open source implementations of all the discussed algorithms.

### 6.2 Learning Hyperparameters and Data Augmentation for Generalization in RL

There has recently been increased interest in testing RL algorithms procedurally generated environments [48, 13]. These typically consist of a set of parameters which are used to generate *levels*, consisting of changes to the observation or layout of the environment, requiring agents to generalize [32, 61]. In this paper we focus on the Procgen environment [13]. Given the recent success of data augmentation in Procgen, we adopt the author's implementation of *data regularized actor critic* or DrAC [57], using a ResNet architecture [22] as originally introduced in IMPALA [19]. As well as the categorical data augmentation, we also tune the regularization coefficient $\alpha_r$ introduced in DrAC, as well as more classically well-studied PPO hyperparameters such as the clip parameter $\epsilon$, the learning rate and entropy exploration coefficient. The bounds for each, as well as the fixed hyperparameters, are given in the Appendix (Section A, Table A.1 and 3). We learn these hyperparameters, as well as the categorical data augmentation *on the fly* in a single training run, starting with random values.

We train a population of $B = 4$ agents, each for 25M steps, with $t_{\text{ready}} = 500k$ steps, thus the explore step is called approximately fifty times. All of the PBT-style methods use the same truncation selection approach, whereby the bottom 25% of agents are replaced with copies of the top 25%. Since we have a population size of $B = 4$ this equates to replacing the worst agent with the best. We use seven Procgen games: BigFish, CaveFlyer, CoinRun, FruitBot, Jumper, Leaper and StarPilot. At the end, each agent is evaluated on 100 train and test levels and we present the test performance from the agent with the best final training performance in each population. Our primary baseline is PB2-Rand, which we note was already shown to improve upon both Hyperband [40, 41] and vanilla BO [8] in [52]. As such, we choose to use our computational resources to run five trials for all seven games rather than by confirming this result. Instead, we do include the original PBT algorithm as a baseline, as well as single agent results from [57]. All of the PBT results can be replicated using our open source codebase, which was designed to make use of our single GPU available and thus is not parallelized.[2] For a parallel version of PB2, see Ray Tune [45, 42].[3]

---

[2]https://github.com/jparkerholder/procgen_autorl
[3]https://docs.ray.io/en/master/tune/examples/pb2_ppo_example.html

Table 2: Test performance for seven Procgen games. † indicates training was conducted for a single agent for 25M timesteps. Hyperparameters were initialized at optimized values, meaning the effective number of timesteps is much higher. Final performance is the average of 100 trials and results were taken from [57]. ‡ indicates training was conducted by a population of four agents for 25M timesteps each, with several hyperparameters initialized at random. Each agent is evaluated for 100 trials on train and test levels and we present the mean test performance of the agent with the best training performance. The "Normalized Returns" are with respect to the PB2-Rand baseline, $\star$ indicates $p < 0.05$ in a t-test over PB2-Rand. Bold = within 1 std of the best mean.

| Environment | PPO† | UCB-DrAC† | PBT‡ | PB2-Rand‡ | PB2-Mult‡ | PB2-Mix‡ |
|---|---|---|---|---|---|---|
| BigFish | $4.0 \pm 1.2$ | $\mathbf{9.7 \pm 1.0}$ | $6.6 \pm 2.4$ | $8.2 \pm 0.1$ | $9.6 \pm 1.7$ | $10.6 \pm 1.9$ |
| CaveFlyer | $5.1 \pm 0.9$ | $\mathbf{5.3 \pm 0.9}$ | $4.4 \pm 1.0$ | $\mathbf{5.8 \pm 1.8}$ | $5.6 \pm 0.8$ | $\mathbf{6.0 \pm 0.7}$ |
| CoinRun | $\mathbf{8.5 \pm 0.5}$ | $\mathbf{8.5 \pm 0.5}$ | $6.8 \pm 1.0$ | $\mathbf{8.1 \pm 0.1}$ | $\mathbf{8.5 \pm 0.4}$ | $\mathbf{8.1 \pm 0.2}$ |
| FruitBot | $26.7 \pm 0.8$ | $28.3 \pm 0.9$ | $25.6 \pm 0.2$ | $\mathbf{29.5 \pm 1.9}$ | $\mathbf{29.6 \pm 0.8}$ | $\mathbf{29.0 \pm 0.6}$ |
| Jumper | $5.8 \pm 0.5$ | $\mathbf{6.4 \pm 0.6}$ | $\mathbf{6.2 \pm 0.3}$ | $5.4 \pm 0.5$ | $5.9 \pm 0.1$ | $\mathbf{6.2 \pm 0.3}$ |
| Leaper | $4.9 \pm 0.7$ | $5.0 \pm 0.3$ | $3.6 \pm 0.6$ | $5.0 \pm 2.4$ | $5.0 \pm 1.4$ | $\mathbf{6.6 \pm 1.9}$ |
| StarPilot | $24.7 \pm 3.4$ | $30.2 \pm 2.8$ | $26.9 \pm 7.9$ | $33.6 \pm 3.1$ | $\mathbf{35.3 \pm 1.5}$ | $\mathbf{36.3 \pm 2.4}$ |
| Normalized Returns (%) | | | $85.6 \pm 22.4$ | $100 \pm 23.8$ | $105.3 \pm 14.6$ | $112.1 \pm 20.4^{\star}$ |

The final results are shown in Table 2, where each agent is evaluated for 100 trials, a common protocol in Procgen [57, 31]. For population based methods, we select the agent with the best *training performance* and show its' test performance. Given the challenge of comparing results, we present an additional metric which is a score normalized by the PB2-Rand performance. For each seed in each game, the score is divided by the mean PB2-Rand score for that given game. We then show the mean of these normalized scores, comprising of 35 results. We show full train/test learning curves in Fig. 7 and 8 in the Appendix. In addition, we note the challenge in comparing aggregate metrics in RL, and as such, we also make use of the recently introduced `rliable` library, designed to provide robust metrics that take into account the uncertainty in experimental results [1]. In Fig.3 we show aggregate performance statistics across all tasks, where we clearly see that PB2-Mix and PB2-Mult outperform PB2-Rand. Our other results confirm what was already known—with limited data ($B = 4$) the original PBT performs poorly (worse than PPO without data augmentation), while PB2-Rand provides large improvements over this.

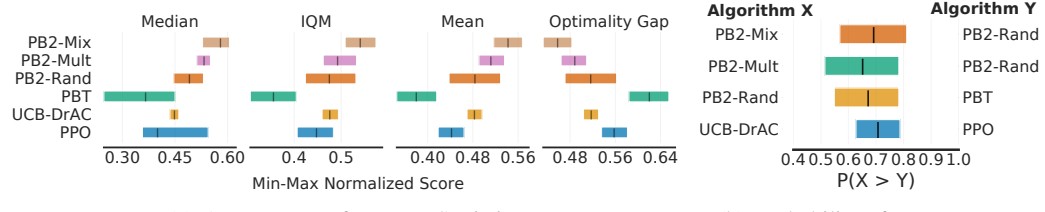

(a) Aggregate Performance Statistics      (b) Probability of Improvement

Figure 3: (a) Aggregate performance statistics across all seven Procgen environments. The authors of [1] recommend IQM. (b) The probability of improvement on a new task from the same distribution.

Despite using a small population size, we are able to train policies that perform competitively with state-of-the-art methods which used a much larger grid search. We see that PB2-Rand significantly outperforms PBT, and furthermore our new approaches modeling categorical variables improve upon PB2-Rand. Notably, we see the strongest performance from PB2-Mix, which is able to exploit the dependence between all variables in a joint kernel. The average gains for PB2-Mix over PB2-Rand are statistically significant, with $p = 0.044$ in Welch's t-test.

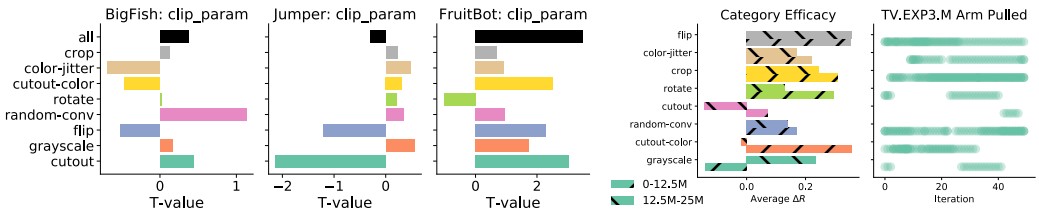

(a) Hyperparameter impact by augmentation type      (b) Augmentation selection efficacy

Figure 4: (a) T-values for univariate regressions predicting reward change given a continuous hyperparameter, conditioned on category. (b) Left: Mean reward change for each category in the BigFish environment, across all experiments, in the first and second half of training. Right: All arms selected by TV.EXP3.M.

**Do we need to model dependence?** To explore the dependence of the parameters in the Procgen setting, we tested relationship between each continuous hyperparameter and the subsequent change in reward ($f_t(.)$). For each *individual* hyperparameter, we *conditioned the data on the category* currently being used, and fit a linear regression model. In Fig. 4.a) we show the t-values for three separate examples, with the full grid in the Appendix (Fig. 9, Section B). As we see, the relationship between continuous variables and learning performance varies depending on the category selected, confirming a dependence in Procgen [72]. In particular, we see for BigFish and Jumper the clip parameter relationship with training performance is heavily dependent on the data augmentation type used. Interestingly we also include an example from FruitBot, where the dependence seems to be weaker as the clip param is positively related with improving policies for all but one category. This is likely the reason for relatively stronger performance for PB2-Rand in FruitBot.

We also investigate the effectiveness of TV.EXP3.M in selecting the data augmentation type. In Fig. 4.b) we show the mean change in reward for each category in the BigFish environment, as well as all the categories selected by the new PB2 variants using TV.EXP3.M. We see that the three most selected augmentations (flip, crop and color-jitter) all lead to positive rewards. Interestingly, we also see that cutout-color is frequently selected at the *beginning*, but not used at all at the end where it no longer improves training, thus demonstrating TV.EXP3.M is able to adapt effectively.

Now we have seen there is clear evidence of dependence between the hyperparameters, we seek to answer this question on the downstream task by training RL agents. We introduce an ablation of our method called "PB2-Indep" which uses TV.EXP3.M to select the data augmentation type but then ignores this information when using the GP to select continuous hyperparameters. We once again evaluate this method with five runs on all seven Procgen games, and show the aggregate results in Fig. 5. Indeed, we see that PB2-Indep does improve upon PB2-Rand, since selecting the best data augmentation should provide performance gains, but we do not see the full benefit of PB2-Mix which is able to select continuous variables conditioned on this information. We see that the probability of improvement for PB2-Mix over PB2-Indep and PB2-Indep over PB2-Rand are both equal (just over $56\%$), which indicates the gain for PB2-Mix comes equally from efficiently selecting the data augmentation type and then subsequently conditioning on the output of that selection when selecting continuous variables. When combined, we get over $69\%$ probability of improvement vs. the baseline.

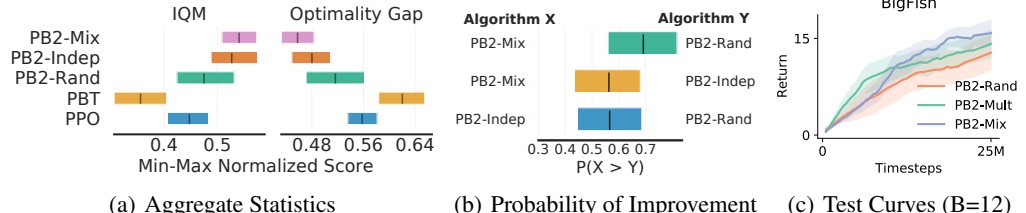

(a) Aggregate Statistics     (b) Probability of Improvement    (c) Test Curves (B=12)

Figure 5: **Ablation Study**:(a) and b) show aggregate performance statistics vs. an ablation "PB2-Indep" which independently (yet efficiently) selects continuous and categorical variables. **Scaling up**: c) shows the test curves with $B = 12$ agents, plots show the mean of three runs with sem shaded.

**Does it scale?** The key benefit of PBT algorithms is the ability to make use of large population sizes with the same wall clock time of a single agent. Thus far, we have only considered the resource constrained setting which is important but not representative of a setting that would be useful for labs with access to more resources. Thus, the question remains: how do these new methods scale? To test this we train with a population size of 12 for PB2-Rand, PB2-Mult and PB2-Mix on the BigFish task. We repeat the experiment for three seeds and report the mean test return from the agent with the best training performance in 5c). As we see, PB2-Mix remains the strongest method, with final performance of $16 \pm 1.15$. This is comparable to the state of the art performance [31, 56]. Since our method is orthogonal to these, it is likely combining them could provide further gains.

## 6.3 Discussion

Our results show the benefit of AutoRL methods for adaptively selecting data augmentation type alongside other RL hyperparameters, with strong results in a set of procedurally generated environments. We believe that with access to more compute, it may be possible to combine our AutoRL framework with ideas from state-of-the-art methods [56, 15, 31] to achieve state-of-the-art results on

the Procgen benchmark. Furthermore, this may be an effective approach to find settings on new PCG environments, when the optimal base configuration is not known.

However, we note there are some limitations in our approach. Firstly, PB2-Mult, which has the strongest theoretical results and performance on the synthetic task, was less impressive in Procgen. This may be due to the continuous/categorical dependence not being strong enough to justify only using a small subset of the data when exploring. We thus believe PB2-Mix is likely the best bet for a novel problem, as it has the ability to trade off between the continuous and categorical kernels. However, optimizing this parameter at each explore step does introduce additional computational complexity, which increases the runtime proportionally to the amount of data. The runtime can be improved by fixing the kernel hyperparameters, using a sliding window of data, or using a faster GP implementation (such as a sparse GP [69]), although in some cases this may not be desirable. Indeed, there is inevitably a trade-off in terms of the desired efficiency of the explore step and the computational cost one is willing to incur. In the extreme setting of cheap evaluations, it may even be preferable to have a larger population of PBT agents, while in the case of Procgen (where each run takes up to 24h on a GPU) it is certainly the opposite, as the explore step takes up a trivial amount of time relative to RL training.

Finally we note another potential area for improvement in our approach: we make decisions in the explore/exploit step based on *training* performance (on changing levels), ignoring generalization ability. This may lead to accidentally removing the best agents on the test set, while it may also lead to inaccuracy in our GP models. One potentially promising approach may be to consider a fixed set of "validation levels", learned using techniques such as *Prioritized Level Replay* [31].

# 7    Conclusion and Future Work

In this paper we expand the capabilities of population based bandits (PB2) by making it possible to efficiently select *both* continuous and categorical hyperparameters. We introduced a new time varying multi-armed bandit algorithm for selecting categorical hyperparameters, and presented a hierarchical approach to subsequently select continuous parameters. We believe this work is an important step in increasing the capability of population-based approaches for AutoRL. This should provide benefits at both ends of the spectrum: it can make RL accessible to a wider audience by significantly lowering the cost of tuning hyperparameters, while also potentially leading to stronger performance for those with larger resources. We also showed the effectiveness of learning both hyperparameters and data augmentation on the fly, for challenging procedurally generated environments. We are not the first to propose population-based methods in this setting [60], but we provided strong evidence that PBT-style hyperparameter tuning can provide significant benefits.

We are particularly excited by many future directions from here. The most natural is to learn more: can we learn policy architectures in the explore step, for example making use of kernels between networks [79]? Can we learn algorithms altogether, extending recent work [49, 12]? We can even consider using our approach to configure environments [31, 16, 30, 62], which may make it possible to jointly learn an environment and algorithm or architecture. Outside of increasing the search space, improvements can be made on the algorithm itself, for example exploring whether it is possible to share experience across agents [20]. We can also consider whether we can benefit from encouraging these agents to be diverse with respect to one another [53, 55]. Finally, we are also curious to see whether recent innovations in other areas of AutoML can be combined in a PBT-style framework [59]. We think the combination of these ideas could lead to dramatic gains in performance for RL agents, ultimately making it possible to go beyond the capability of human-designed configurations.

# Acknowledgements

The experiments in this paper were conducted using AWS. SD would like to thank the Rhodes Trust for supporting this research. The authors would like to thank Roberta Raileanu for providing open source code and both Roberta and Minqi Jiang for discussion on reporting for Procgen results, as well as Xingyou Song and Yingjie Miao for discussion around AutoRL. Finally, this work was improved thanks to constructive feedback from anonymous reviewers.

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
