# A Implementation Details

## A.1 Hyperparameter Ranges

**RL Hyperparameters**: Below are the continuous hyperparameters optimized as well as the fixed parameters, which were all taken from [57]. For the categorical variables, we use the data augmentations used in [57, 37, 35]: *crop, grayscale, cutout, cutout-color, flip, rotate, random convolution and color-jitter*.

Table 3: DrAC Learned Hyperparameters

| Parameter | Value |
|---|---|
| PPO Clip ($\epsilon$) | $[0.01, 0.5]$ |
| Learning Rate | $[10^{-5}, 10^{-3}]$ |
| Entropy Coeff | $[0, 0.2]$ |
| Regularization Parameter ($\alpha_r$) | $[0.01, 0.5]$ |

Table 4: DrAC Fixed Hyperparameters

| Parameter | Value |
|---|---|
| $\gamma$ | 0.999 |
| $\lambda$ | 0.95 |
| # timesteps per rollout | 256 |
| # epochs per rollout | 3 |
| # minibatches per epoch | 8 |
| optimizer | Adam |

**PB2 Hyperparameters**: We use the same UCB hyperparameters as in [52], which come from [7]. Concretely, we set $c1 = 0.2$, $c2 = 0.4$. We believe performance may be increased by selecting these parameters more carefully. However, it is promising to see that this fine tuning is not crucial for performance.

## A.2 Environment Details

We used five games from the Procgen environment [13]. We chose these games as there appeared to be a greater degree of variance in the optimal data augmentation in [57]. All protocols for the environment are from [57], as we followed the author's open source implementation for agent training[4], and only varied the hyperparameters and data augmentation chosen.

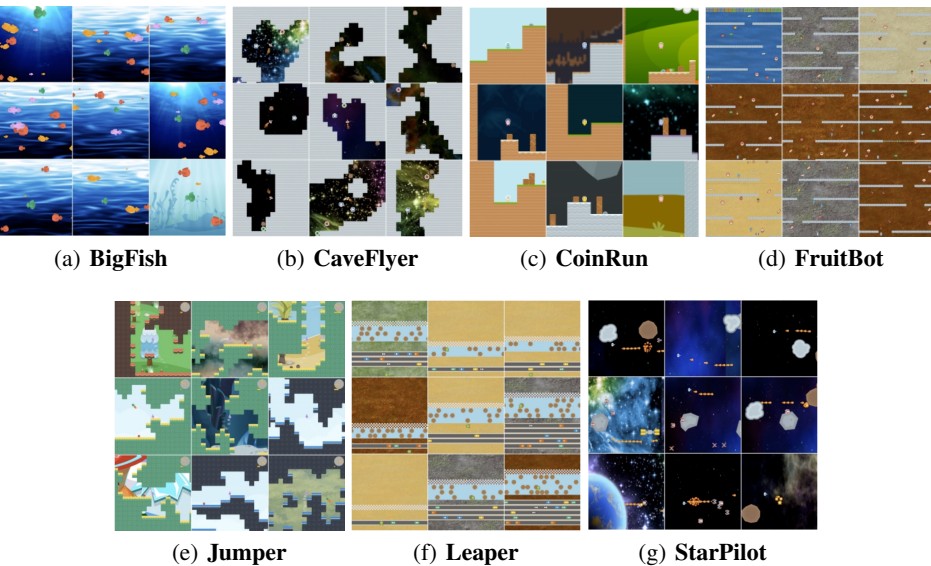

(a) **BigFish**  (b) **CaveFlyer**  (c) **CoinRun**  (d) **FruitBot**

(e) **Jumper**  (f) **Leaper**  (g) **StarPilot**

Figure 6: Example levels from the seven games considered from [13]

---

[4]see: https://github.com/rraileanu/auto-drac

### A.3 Infrastructure Details

Each trial was run a single GPU, taking around three days to train all four agents. This was due to the nature of our existing compute infrastructure and could likely be made significantly faster with parallelization. The existing code is thus not optimized to work in a distributed fashion, however it will be open sourced in a similar structure to the open source PB2-Rand implementation (part of the Ray library [45]). Since we followed the open source version of PB2-Rand, it is a trivial extension to include PB2-Mult and PB2-Mix, using new kernels provided both in the supplementary material (as part of the research code) or the synthetic function notebook (see the link in Section 6).

## B   Additional Results

Here we show the test learning curves in Fig. 7 where we see particularly strong improvement in BigFish and CaveFlyer. Notably, PBT performs poorly in many of the tasks.

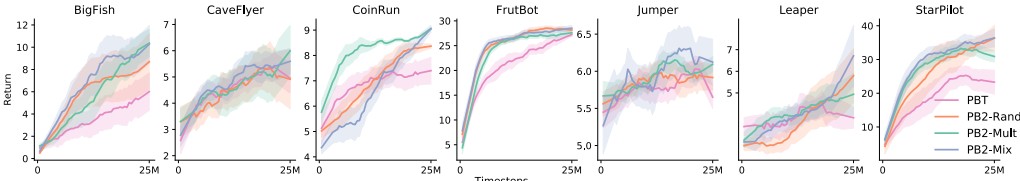

Figure 7: **Test Performance**: Learning curves for seven Procgen games. Plots show the mean ±1sem for the test performance of the best training agents in the population. Results are averaged over 5 seeds.

Table 5: **Train** performance for seven Procgen games. † indicates training was conducted for a single agent for 25M timesteps. Hyperparameters were initialized at optimized values, meaning the effective number of timesteps is much higher. Final performance is the average of 100 trials and results were taken from [57]. ‡ indicates training was conducted by a population of four agents for 25M timesteps each, with several hyperparameters initialized at random. Each agent is evaluated for 100 trials on train and test levels and we present the mean train performance of the agent with the best training performance.

| Environment | PPO† | UCB-DrAC† | PBT‡ | PB2-Rand‡ | PB2-Mult‡ | PB2-Mix‡ |
|---|---|---|---|---|---|---|
| BigFish | $8.9 \pm 1.5$ | $13.2 \pm 2.2$ | $12.0 \pm 3.3$ | $20.8 \pm 1.8$ | $18.2 \pm 2.0$ | $17.1 \pm 2.9$ |
| CaveFlyer | $6.8 \pm 0.6$ | $5.7 \pm 0.6$ | $7.2 \pm 0.4$ | $7.3 \pm 1.7$ | $7.0 \pm 1.3$ | $7.5 \pm 0.9$ |
| CoinRun | $9.3 \pm 0.3$ | $9.5 \pm 0.3$ | $8.2 \pm 0.8$ | $10.0 \pm 0$ | $9.9 \pm 0.2$ | $9.6 \pm 0.2$ |
| FruitBot | $29.1 \pm 1.1$ | $29.5 \pm 1.2$ | $27.4 \pm 0.2$ | $32.2 \pm 0.7$ | $30.5 \pm 1.4$ | $30.9 \pm 1.2$ |
| Jumper | $8.3 \pm 0.4$ | $8.1 \pm 0.7$ | $8.3 \pm 0.2$ | $9.0 \pm 0.2$ | $8.9 \pm 0.1$ | $9.2 \pm 0.5$ |
| Leaper | $5.5 \pm 0.4$ | $5.3 \pm 0.5$ | $4.1 \pm 0.5$ | $6.9 \pm 2.0$ | $5.6 \pm 2.1$ | $7.1 \pm 2.3$ |
| StarPilot | $29.8 \pm 2.3$ | $35.3 \pm 2.2$ | $33.6 \pm 7.4$ | $44.1 \pm 2.7$ | $40.3 \pm 1.3$ | $41.8 \pm 2.7$ |

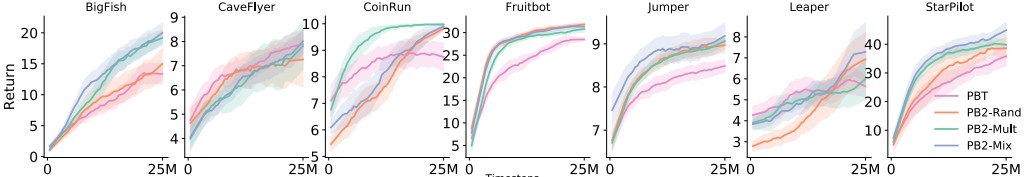

Figure 8: **Train Performance**: Learning curves for all seven Procgen games. Plots show the mean ±1sem for the best training performance within the population. Results are averaged over 5 seeds.

In Table 5 we show training performance for the agents shown in the main paper. Interestingly, we see strong training performance for PB2-Rand, indicating a larger generalization gap. We also show training learning curves for all algorithms in Fig. 8, where we once again report the best agent at each timestep.

**Is there dependence between data augmentation and hyperparameters in Procgen?** Next we consider the dependence between the hyperparameters. For each environment, we fit a linear regression model predicting the change in reward for 500k steps of training (i.e. one step of PB2), with the independent variable being **one** of the continuous hyperparameters. Colors correspond to the setting where we condition on an individual category (by creating a subset), while black corresponds to using the entire dataset. We use the data from all trials and show the results in In Fig. 9.

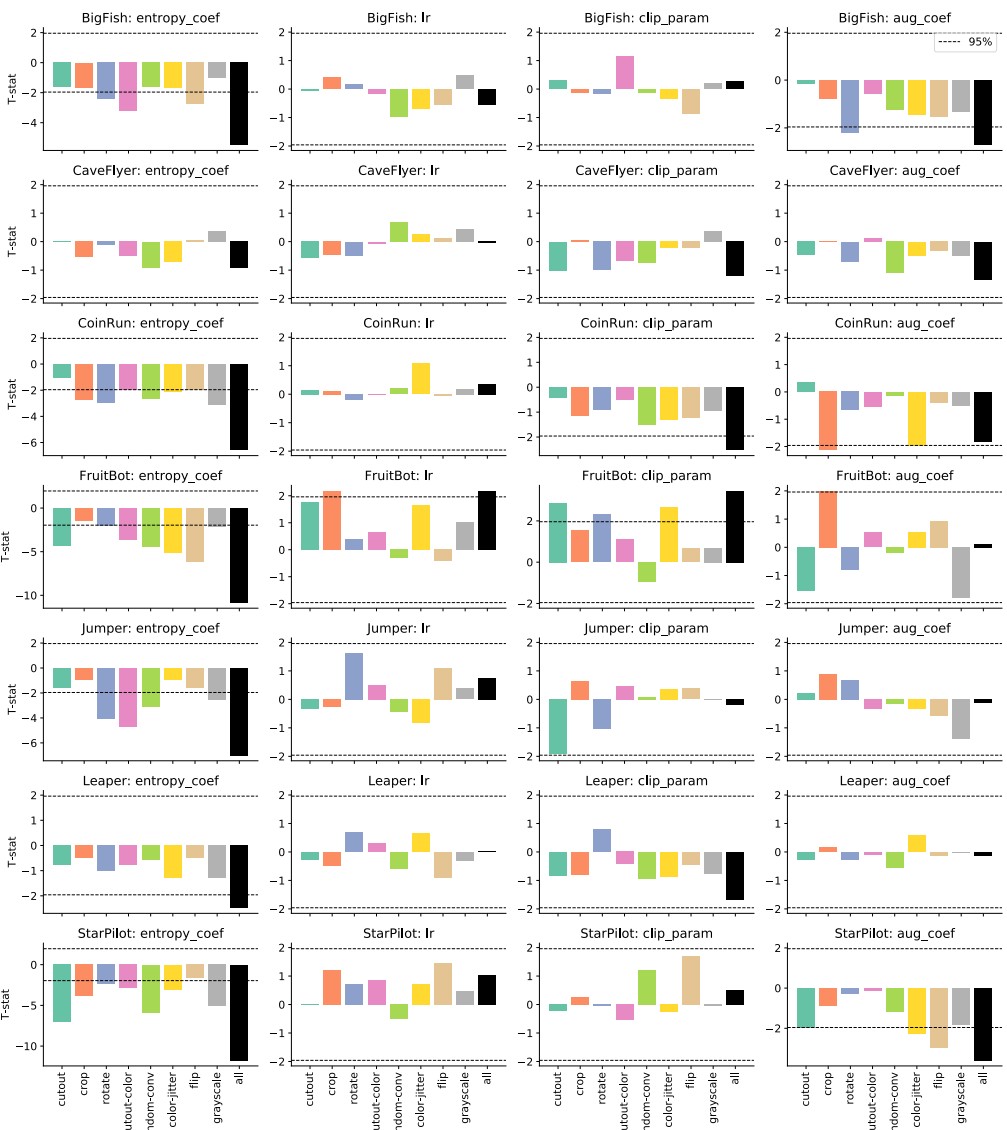

Figure 9: t-statistic for the independent variable (one continuous hyperparameter, in the title of each plot) in predicting the change in reward for the given environment. Blue columns represent the data being from one. The dotted line indicates $p < 0.05$.

As we see, in some cases the category does not have any impact on the relationship, for example the entropy coefficient for CoinRun: an increase in entropy appears to be negatively related with training performance across all augmentations. However, for some settings such as the clip parameter in BigFish, we see large swings in the t-stat depending on category. We even see some settings where the aggregate relationship (grey) is very small, but conditioning on the category makes it worthwhile to tune the hyperparameter, for example the learning rate for CaveFlyer. In this case it may be challenging for PB2-Rand to achieve gains from tuning the learning rate. We note this analysis is purely illustrative and has many assumptions, in particular 1) we use a linear univariate model 2) we ignore the time-varying component.

**Learned Schedules** The next pages contain learned schedules from all agents trained by our algorithms and baselines. In each case we show all agents for all seeds, with blue corresponding to a config that contributed to the final best (training) agent. The most notable observation across all environments is the stark contrast between PBT and PB2-based methods. PBT clearly increments parameters, and gradually shifts during training, while PB2 rapidly explores the boundaries. We note in some cases that PB2-Mult never re-explores the middle regions, which we hypothesize is due to the lack of data when creating separate models for each of the eight categories.

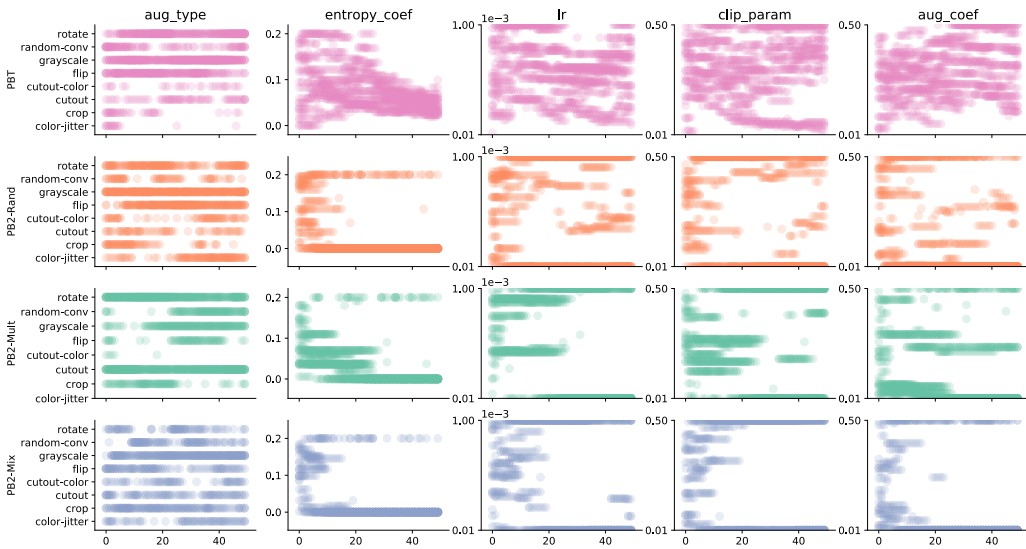

Figure 10: Learned schedules for BigFish by algorithm, for all seeds. Each point corresponds to one agent.

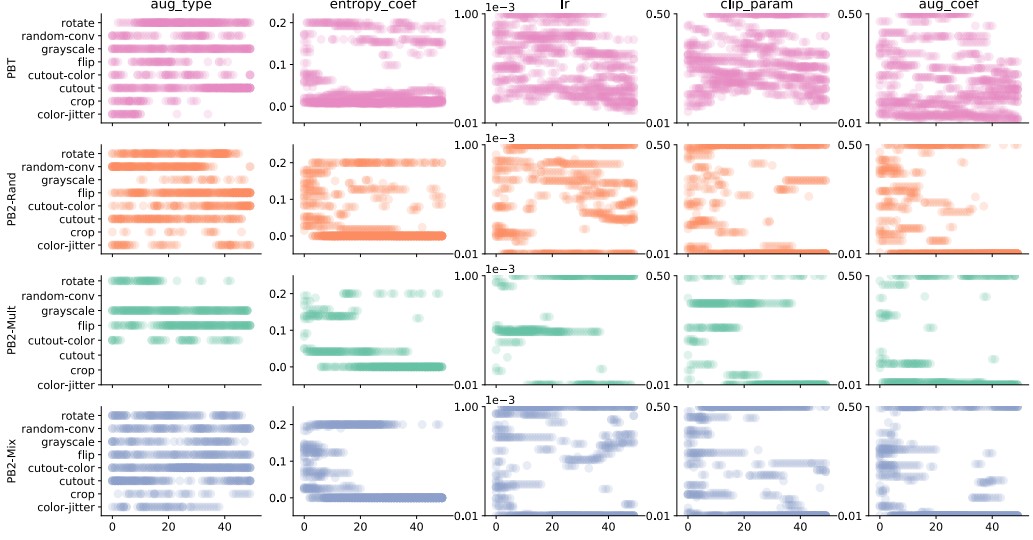

Figure 11: Learned schedules for CaveFlyer by algorithm, for all seeds. Each point corresponds to one agent.

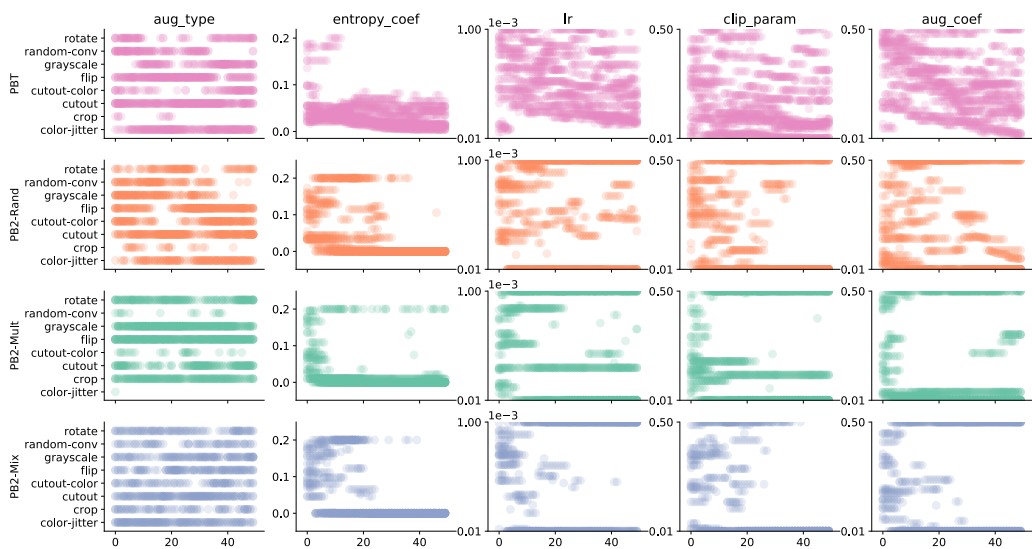

Figure 12: Learned schedules for CoinRun by algorithm, for all seeds. Each point corresponds to one agent.

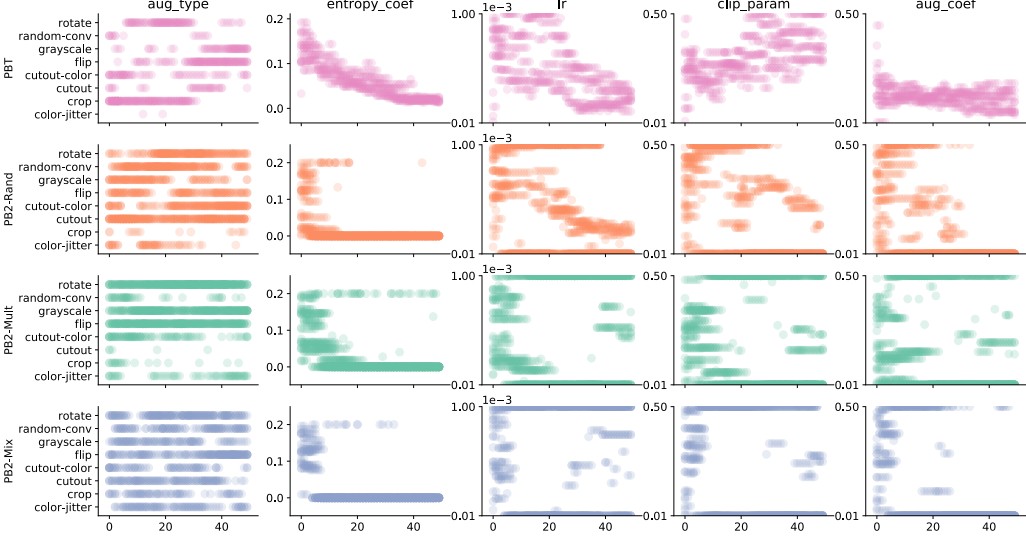

Figure 13: Learned schedules for FruitBot by algorithm, for all seeds. Each point corresponds to one agent.

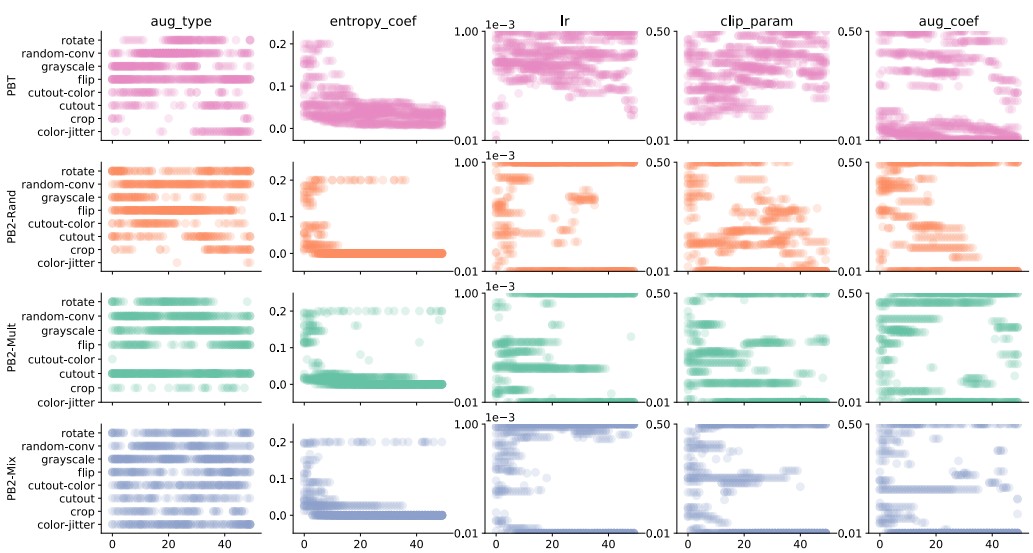

Figure 14: Learned schedules for Jumper by algorithm, for all seeds. Each point corresponds to one agent.

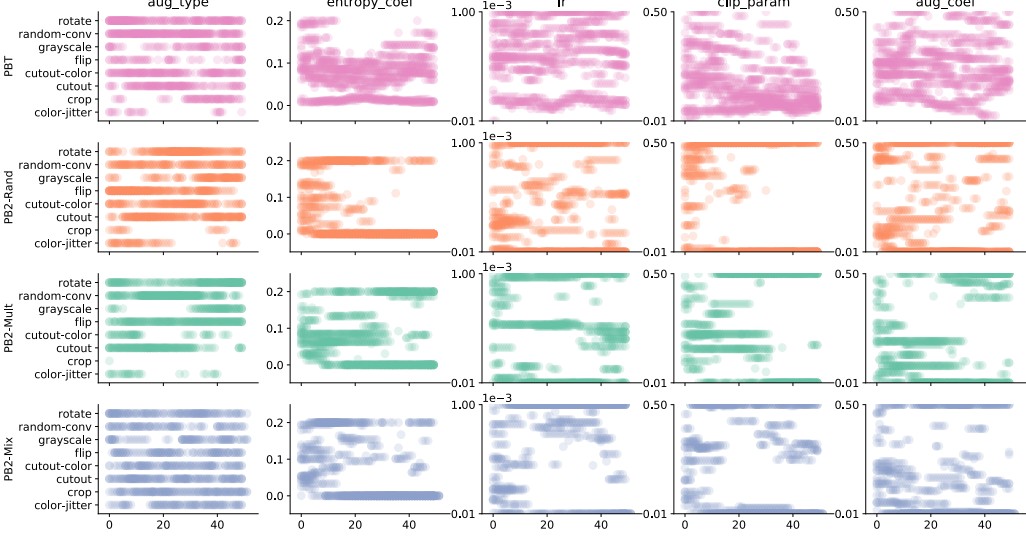

Figure 15: Learned schedules for Leaper by algorithm, for all seeds. Each point corresponds to one agent.

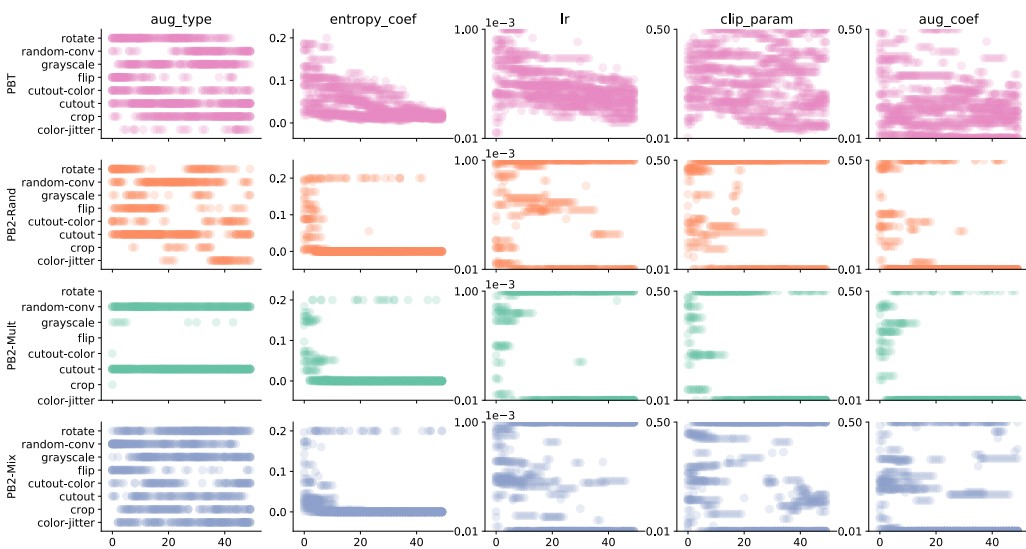

Figure 16: Learned schedules for StarPilot by algorithm, for all seeds. Each point corresponds to one agent.

# C  Additional Background

## C.1  DrAC

In recent times, there has been increased interest in *generalization* of RL agents, after multiple works showed many RL agents are simply overfitting to a single deterministic training environment [51, 14]. One approach to produce more generalizable agents is data augmentation, which has recently shown impressive results [82, 35, 37, 38]. To formalize the data augmentation step, we follow [35] and define an optimality-invariant state transformation $f : \mathcal{S} \times \mathcal{H} \to \mathcal{S}$ as a mapping that preserves both the policy $\pi$ and value function $V$, i.e. $V(s) = V(f(s,\nu))$ and $\pi(a|s) = \pi(a|f(s,\nu)), \forall s \in \mathcal{S}, \nu \in \mathcal{H}$, where $\nu$ are parameters of $f(.)$ drawn from the set of possible parameters $\mathcal{H}$. In this work we focus on the formulation from [57] who propose additional loss terms for regularizing the policy and value function:

$$G_\pi = \text{KL}[\pi_\theta(a|f(s,\nu))|\pi(a|s)], \tag{6}$$

$$G_V = (V_\theta(f(s,\nu)) - V(s))^2. \tag{7}$$

Combining the PPO objective with $G_\pi$ and $G_V$ produces the *data-regularized actor critic* or DrAC objective as follows:

$$\mathcal{L}_{\text{DrAC}}(\theta) = \mathcal{L}_{\text{PPO}}(\theta) - \alpha_r(G_\pi + G_V) \tag{8}$$

where $\alpha_r$ is the weight of the regularization term, another hyperparameter to consider. The results in [57] show that learning which data augmentation function $f$ to use (from a fixed set) and grid searching over the $\alpha_r$ parameter can achieve new state-of-the-art results in challenging Procgen environments.

Table 6: Notation used in the theoretical analysis.

| Variable | Domain | Meaning |
|:---:|:---:|:---:|
| $C$ | $\mathcal{N}$ | number of categories (number of arms) |
| $T$ | $\mathcal{N}$ | maximum number of bandit update (the number of $t_{\text{ready}}$ in PB2) |
| $V$ | $\mathcal{N}$ | number of time segments, how many times the function has been shifted |
| $B$ | $\mathcal{N}$ | batch size (number of parallel agents) |
| $S_t$ | $list$ | a list of $B$ selected categories $[c_{t,1}, c_{t,2}, ..., c_{t,B}]$ |
| $e$ | $\mathcal{R}^+$ | this is Euler's number 2.71828 |
| $[p_t^1, ... p_t^C]$ | $list$ | probability vector at iteration $t$ and category $c = 1....C$ |
| $w_c$ | $\mathcal{R}$ | weight for categorical $c$ (or arm $c$) |
| $W_t = \sum_{c=1}^{C} w_c$ | $\mathcal{R}^+$ | sum of the weight vector at iteration $t$ |
| $c_{t,b} \in \{1...C\}$ | $\mathcal{N}$ | a selection at iteration $t$ at agent $b$ |
| $c_t = [c_{t,1}, ..., c_{t,B}]$ | $list$ | a batch of $B$ categorical selections at iteration $t$ |
| $c_t^* \in \{1...C\}$ | $list$ | an optimal selection at iteration $t$ |
| $A_v^*$ | $list$ | a list of $B$ elements taking the (same) optimal selection $c_t^*$ at iteration/segment $v$ |
| $g_t(c)$ | $[0,1]$ | a gain (reward) occurred at iteration $t$ by pulling an arm $c$ |
| $\hat{g}_t(c) = \frac{g_t(c)}{p_t^c + \gamma} \mathbb{I}(c = h_t)$ | $\mathcal{R}^+$ | a normalized gain |
| $\alpha = \frac{1}{T}, \eta = 2\gamma = \sqrt{\frac{2\ln C}{CT}}$ | $\mathcal{R}^+$ | hyperparameters, set by Theorem 1 |
| $\gamma = \sqrt{\frac{\ln C}{2CT}} \in [0,1]$ | $\mathcal{R}^+$ | is the exploration parameter |
| $G_T(c) = \sum_{t=1}^{T} g_t(c)$ | $\mathcal{R}^+$ | a total gain if we select an arm $c$ entirely |

# D  Theoretical Results

We derive the theoretical proofs presented in the main paper.

## D.1  Time-varying EXP3 Multiple play (TV.EXP3.M)

We present a new algorithm for parallel (or multiple play) multi-armed bandits in the time-varying setting with adversarial feedback. Particularly, we extend the multiple play EXP3 algorithm (or EXP3.M) [78], to the time-varying setting where the unknown reward distribution of each arm can change arbitrarily, but the total number of change points is no more than $V$ times. We refer to Table 6 for the notations used in the proofs.

We summarize the TV.EXP3.M in Algorithm 5 – this is complementary to the brief Algorithm 2 in the main paper. In Algorithm 5, we maintain a set of probability vectors (step 9) for each arm which will be specified and weighted in the optimal way derived by the theory. Then, at each round we select a batch of $B$ arms for parallel evaluations (step 10), observe the reward (step 11) and update the model (step 12,13,14). The normalization steps 3-7 are to prevent from being biased toward a single arm which performs overwhelmingly well, thus overly exploiting.

**Theorem 3.** *(Theorem 1 in the main paper) Let $T > 0$, $C > 0$, set $\alpha = \frac{1}{T}$ and $\gamma = \min\left\{1, \sqrt{\frac{C\ln(C/B)}{(e-1)BT}}\right\}$, we assume the reward distributions to change at arbitrary time instants, but the total number of change points is no more than $V$ times. The expected regret gained by TV.EXP3.M in a batch satisfies the following sublinear regret bound*

$$\mathbb{E}\left[R_{TB}\right] \leq [1 + e + V]\sqrt{(e-1)\frac{CT}{B}\ln\frac{CT}{B}}.$$

*Proof.* We refer to Table 6 for notations. We follow the proof technique presented in [3] and [78] to derive the regret bound. Let $W_t = \sum_{c=1}^{C} \omega_t^c$ and using step 9 in Algorithm 5, we have $\frac{\frac{p_t^c}{B} - \frac{\gamma}{C}}{1 - \gamma} = \frac{\omega_t^c}{W_t}$.

This equation will be used below.

$$\frac{W_{t+1}}{W_t} = \frac{\sum_{c=1}^{C} \omega_{t+1}^c}{W_t} = \frac{\sum_{c \notin S_t(0)} \omega_{t+1}^c}{W_t} + \frac{\sum_{c \in S_t(0)} \omega_{t+1}^c}{W_t} + \frac{\sum_{c=1}^{C} \frac{e\alpha W_t}{C}}{W_t}$$

$$= \sum_{c \notin S_t(0)} \frac{\omega_t^c}{W_t} \times \exp\left(\frac{B\gamma}{C} \hat{g}_t(c)\right) + \frac{\sum_{c \in S_t(0)} \omega_t^c}{W_t} + e\alpha$$

$$\leq e\alpha + \sum_{c \notin S_t(0)} \frac{\omega_t^c}{W_t}\left[1 + \frac{B\gamma}{C}\hat{g}_t(c) + (e-2)\left(\frac{B\gamma}{C}\right)^2 \hat{g}_t^2(c)\right] + \frac{\sum_{c \in S_t(0)} \omega_{t+1}^c}{W_t} \quad \text{by } e^x \leq 1 + x + (e-2)x^2$$

$$= e\alpha + \underbrace{\sum_{c \notin S_t(0)} \frac{\omega_t^c}{W_t} + \sum_{c \in S_t(0)} \frac{\omega_{t+1}^c}{W_t}}_{1} + \sum_{c \notin S_t(0)} \frac{\frac{p_t^c}{B} - \frac{\gamma}{C}}{1-\gamma}\left[\frac{B\gamma}{C}\hat{g}_t(c) + (e-2)\left(\frac{B\gamma}{C}\right)^2 \hat{g}_t^2(c)\right]$$

$$\leq e\alpha + 1 + \frac{B\gamma}{C(1-\gamma)} \sum_{c \notin S_t(0)}\left(\frac{p_t^c}{B} - \frac{\gamma}{C}\right)\hat{g}_t(c) + \frac{(e-2)}{1-\gamma}\left(\frac{B\gamma}{C}\right)^2 \sum_{c \notin S_t(0)}\left(\frac{p_t^c}{B} - \frac{\gamma}{C}\right)\hat{g}_t^2(c)$$

$$= e\alpha + 1 + \frac{\gamma}{C(1-\gamma)} \sum_{c \notin S_t(0)} p_t^c \hat{g}_t(c) - \frac{B\gamma^2}{C^2(1-\gamma)} \sum_{c \notin S_t(0)} \hat{g}_t(c)$$

$$+ \frac{(e-2)\gamma^2 B}{(1-\gamma)C^2} \sum_{c \notin S_t(0)} p_t^c \hat{g}_t^2(c) - \frac{(e-2)\gamma^3 B^2}{(1-\gamma)C^2} \sum_{c \notin S_t(0)} \hat{g}_t^2(c)$$

$$\leq e\alpha + 1 + \frac{\gamma}{C(1-\gamma)} \sum_{c \notin S_t(0)} p_t^c \hat{g}_t(c) + \frac{(e-2)\gamma^2 B}{(1-\gamma)C^2} \sum_{c \notin S_t(0)} p_t^c \hat{g}_t^2(c)$$

$$\leq e\alpha + 1 + \frac{\gamma}{C(1-\gamma)} \sum_{c \in S_t - S_t(0)} g_t(c) + \frac{(e-2)\gamma^2 B}{(1-\gamma)C^2} \sum_{c \notin S_t(0)} \hat{g}_t(c) \quad \text{by } x^2 \leq x, \forall x \in [0,1]$$

$$\frac{W_{t+1}}{W_t} \leq e\alpha + 1 + \frac{\gamma}{C(1-\gamma)} \sum_{c \in S_t - S_t(0)} g_t(c) + \frac{(e-2)\gamma^2 B}{(1-\gamma)C^2} \sum_{c \in [C]} \hat{g}_t(c). \tag{9}$$

To demonstrate the time-varying property in our bandit problem, we assume the reward distributions change at arbitrary time instants, but the total number of change points is no more than $V$ times. We split the sequence of total decisions $T$ into $V$ segments such that within each segment we have the *time-invariant* reward function.

We can write $V$ segments as $[T_1, ...T_2), [T_2, ...., T_3), [T_V, ..., T_{V+1})$ where $T_v$ indicates the starting index of the $v$-th segment, $T_{v+1}$ is the ending index of the $v$-th segment which is also the starting index of the $v+1$-th segment – using the same notation in EXP3.S [3]. Similarly, we can write the optimal sequence $\underbrace{[c_{T_1}^*, ...c_{T_2}^*)}_{=c_{T_1}^*}, \underbrace{[c_{T_2}^*, ...., c_{T_3}^*)}_{=c_{T_2}^*}, \underbrace{[c_{T_V}^*, ..., c_{T_{V+1}}^*)}_{=c_{T_V}^*}$ where the reward function does not change in each segment, thus takes the same optimal choice $c_{T_v}^*$.

We consider an arbitrary segment $v$ and denote the length $\Delta_v = T_{v+1} - T_v$. Furthermore, let define the cumulative gain (or reward) achieved by using TV.EXP3.M strategy within a segment $v$ that the indices are ranging from $T_v$ to $T_{v+1} - 1$

$$G_{TV.EXP3.M}(v) = \sum_{t=T_v}^{T_{v+1}-1} \sum_{c \in S_t} g_t(c_t = c).$$

Taking ln of Eqn. (9), we get

$$\ln \frac{W_{t+1}}{W_t} \leq \ln\left(e\alpha + 1 + \frac{\gamma/C}{1-\gamma} \sum_{c \in S_t - S_t(0)} g_t(c) + \frac{(e-2)\left(\frac{\gamma}{C}\right)^2 B}{1-\gamma} \sum_{c=1}^{C} \hat{g}_t(c)\right)$$

$$\leq e\alpha + \frac{\gamma/C}{1-\gamma} \sum_{c \in S_t - S_t(0)} g_t(c) + \frac{(e-2)\left(\frac{\gamma}{C}\right)^2 B}{1-\gamma} \sum_{c=1}^{C} \hat{g}_t(c) \quad \text{by } 1 + a \leq e^a.$$

Summing over all indices $t = T_v, ...., T_{v+1} - 1$ within a segment $v$-th:

$$\ln W_{T_{v+1}} - \ln W_{T_v} \le \Delta_v e\alpha + \frac{\gamma/C}{1-\gamma} \sum_{t=T_v}^{T_{v+1}-1} \sum_{c\in S_t - S_t(0)} g_t(c) + \frac{(e-2)\left(\frac{\gamma}{C}\right)^2 B}{1-\gamma} \sum_{t=T_v}^{T_{v+1}-1} \sum_{c=1}^{C} \hat{g}_t(c)$$

$$\le \Delta_v e\alpha + \frac{\gamma/C}{1-\gamma} \sum_{t=T_v}^{T_{v+1}-1} \sum_{c\in S_t - S_t(0)} g_t(c) + \frac{(e-2)\left(\frac{\gamma}{C}\right)^2 B}{1-\gamma} \sum_{t=T_v}^{T_{v+1}-1} \sum_{c=1}^{C} \hat{g}_t(c).$$

Let $j = c^*_{T_v} = ... = c^*_{T_{v+1}-1}$ be the optimal choice in the sequence $v$, we have $\omega_j^{(T_{v+1})} = \omega_j^{(T_{v+1}-1)} \times \exp\left(\frac{B\gamma}{C}\hat{g}_t(j)\right) + \frac{e\alpha}{C}W_{T_{v+1}-1}, \forall j \notin S_{T_{v+1}-1}(0)$ and $\omega_j^{(T_{v+1})} = \omega_j^{(T_{v+1}-1)}, \forall j \in S_{T_{v+1}-1}(0)$

$$\omega_j^{(T_{v+1})} = \omega_j^{(T_{v+1}-1)} \times \exp\left(\frac{B\gamma}{C}\hat{g}_t(j)\right) + \frac{e\alpha}{C}W_{T_{v+1}-1}$$

$$\ge \omega_j^{(T_{v+1}-1)} \times \exp\left(\frac{B\gamma}{C}\hat{g}_t(j)\right)$$

$$\ge \omega_j^{(T_{v+1}-2)} \times \exp\left(\frac{B\gamma}{C}\hat{g}_t(j)\right) \exp\left(\frac{B\gamma}{C}\hat{g}_{(T_{v+1}-2)}(j)\right)$$

$$\ge \omega_j^{(T_v)} \times \exp\left(\frac{B\gamma}{C} \sum_{t=T_v | t:j\notin S_t(0)}^{T_{v+1}-1} \hat{g}_t(j)\right)$$

$$\ge \frac{e\alpha}{C}W_{T_v} \times \exp\left(\frac{B\gamma}{C} \sum_{t=T_v | t:j\notin S_t(0)}^{T_{v+1}-1} \hat{g}_t(j)\right)$$

$$\ge \frac{\alpha}{C}W_{T_v} \times \exp\left(\frac{B\gamma}{C} \sum_{t=T_v | t:j\notin S_t(0)}^{T_{v+1}-1} \hat{g}_t(j)\right).$$

We now consider the lower bound by taking the optimal (batch) set $A_v^* \subset [C]$ for $B$ elements with the maximum total of reward within the segment $v$: $\sum_{c\in A_v^*} \sum_{t=1}^{T} g_t(c)$. Since we have the fact that $\sum_{j\in A_v^*} \omega_j^{(T_{v+1})} \ge B\left(\prod_{j\in A_v^*} \omega_j^{(T_{v+1})}\right)^{1/B}$ by Cauchy Swatchz inequality, we continue the above formula

$$\ln\left(W_{T_{v+1}-1}\right) - \ln W_{T_v} \ge \ln B + \frac{1}{B}\sum_{j\in A_v^*} \ln \omega_j^{(T_{v+1})} - \ln W_{T_v}$$

$$= \ln B + \frac{1}{B}\sum_{j\in A_v^*} \ln \frac{\alpha}{C}W_{T_v} + \frac{1}{B}\sum_{j\in A_v^*} \frac{B\gamma}{C} \sum_{t=T_v | t:j\notin S_t(0)}^{T_{v+1}-1} \hat{g}_t(j) - \ln W_{T_v}$$

$$= \ln B + \frac{1}{B}\sum_{j\in A_v^*} \ln \frac{\alpha}{C} + \sum_{j\in A_v^*} \frac{\gamma}{C} \sum_{t=T_v | t:j\notin S_t(0)}^{T_{v+1}-1} \hat{g}_t(j)$$

$$= \ln \frac{B\alpha}{C} + \sum_{j\in A_v^*} \frac{\gamma}{C} \sum_{t=T_v | t:j\notin S_t(0)}^{T_{v+1}-1} \hat{g}_t(j).$$

Combining both the lower bound and upper bound, we get

$$\ln \frac{B\alpha}{C} + \sum_{j\in A_v^*} \frac{\gamma}{C} \sum_{t=T_v | t:j\notin S_t(0)}^{T_{v+1}-1} \hat{g}_t(j) \le \Delta_v e\alpha + \frac{\gamma/C}{1-\gamma} \sum_{t=T_v}^{T_{v+1}-1} \sum_{c\in S_t - S_t(0)} g_t(c)$$

$$+ \frac{(e-2)\left(\frac{\gamma}{C}\right)^2 B}{1-\gamma} \sum_{t=T_v}^{T_{v+1}-1} \sum_{c=1}^{C} \hat{g}_t(c).$$

Summing over all segments $v = 1....V$

$$V \ln \frac{B\alpha}{C} + \sum_{v=1}^{V} \sum_{j \in A_v^*} \frac{\gamma}{C} \sum_{t=T_v|t:j \notin S_t(0)}^{T_{v+1}-1} \hat{g}_t(j) \leq Te\alpha + \frac{\gamma/C}{1-\gamma} \sum_{v=1}^{V} \sum_{t=T_v}^{T_{v+1}-1} \sum_{c \in S_t - S_t(0)} g_t(c)$$

$$+ \frac{(e-2)\left(\frac{\gamma}{C}\right)^2 B}{1-\gamma} \sum_{v=1}^{V} \sum_{t=T_v}^{T_{v+1}-1} \sum_{c=1}^{C} \hat{g}_t(c)$$

$$\frac{VC}{\gamma} \ln \frac{B\alpha}{C} + \sum_{v=1}^{V} \sum_{j \in A_v^*} \sum_{t=T_v|t:j \notin S_t(0)}^{T_{v+1}-1} \hat{g}_t(j) \leq \frac{C}{\gamma} Te\alpha + \frac{1}{1-\gamma} \sum_{v=1}^{V} \sum_{t=T_v}^{T_{v+1}-1} \sum_{c \in S_t - S_t(0)} g_t(c)$$

$$+ \frac{(e-2)\left(\frac{\gamma B}{C}\right)}{1-\gamma} \sum_{v=1}^{V} \sum_{t=T_v}^{T_{v+1}-1} \sum_{c=1}^{C} \hat{g}_t(c)$$

$$\sum_{v=1}^{V} \sum_{t=T_v|j \in S_t(0)}^{T_{v+1}-1} \sum_{j \in A_v^*} g_t(j) + \sum_{v=1}^{V} \sum_{j \in A_v^*} \sum_{t=T_v|j \notin S_t(0)}^{T_{v+1}-1} \hat{g}_t(j) \leq \frac{C}{\gamma} Te\alpha + \frac{1}{1-\gamma} \sum_{v=1}^{V} \sum_{t=T_v}^{T_{v+1}-1} \sum_{c \in S_t(0)} g_t(c)$$

$$+ \frac{1}{1-\gamma} \sum_{v=1}^{V} \sum_{t=T_v}^{T_{v+1}-1} \sum_{c \in S_t - S_t(0)} g_t(c)$$

$$+ \frac{(e-2)\left(\frac{\gamma B}{C}\right)}{1-\gamma} \sum_{v=1}^{V} \sum_{t=T_v}^{T_{v+1}-1} \sum_{c=1}^{C} \hat{g}_t(c)$$

$$- \frac{VC}{\gamma} \ln \frac{B\alpha}{C}. \tag{10}$$

Let us denote the optimal gain over all iterations and all parallel agents $G_{TB}^* = \sum_{v=1}^{V} \sum_{t=T_v}^{T_{v+1}-1} \sum_{j \in A_v^*} g_t(j)$. We also denote the cumulative gain achieved by using TV.EXP3.M algorithms $G_{\text{TV.EXP3.M}} = \sum_{v=1}^{V} \sum_{t=T_v}^{T_{v+1}-1} \sum_{c \in S_t} g_t(c)$. Then, we continue Eqn. (10) as

$$G_{TB}^* \leq \frac{C}{\gamma} Te\alpha + \frac{1}{1-\gamma} G_{\text{TV.EXP3.M}} + \frac{(e-2)\left(\frac{\gamma B}{C}\right)}{1-\gamma} \sum_{v=1}^{V} \sum_{t=T_v}^{T_{v+1}-1} \sum_{c=1}^{C} \hat{g}_t(c) - \frac{VC}{\gamma} \ln \frac{B\alpha}{C} \tag{11}$$

where the number of element in a batch $|S_t| = |A_j^*| = B$. Let take expectation both sides of Eqn. (11), we have $\mathbb{E}\left[\hat{g}_t(c) \mid S_1, S_2, ..., S_{i-1}\right] = g_t(c)$ from the fact that DepRound [21] selects action $c$ with probability $p_c(t)$, we obtain

$$G_{TB}^* \leq \frac{1}{(1-\gamma)} \mathbb{E}\left[G_{\text{TV.EXP3.M}}\right] + \frac{(e-2)\left(\frac{\gamma B}{C}\right)}{1-\gamma} \sum_{v=1}^{V} \sum_{t=T_v}^{T_{v+1}-1} \sum_{c=1}^{C} g_t(c) + \frac{C}{\gamma} Te\alpha - \frac{VC}{\gamma} \ln \frac{B\alpha}{C}.$$

We finally have

$$(1-\gamma)G_{TB}^* \leq \mathbb{E}\left[G_{\text{TV.EXP3.M}}\right] + (e-2)\left(\frac{\gamma B}{C}\right) \frac{C}{B} G_{TB}^* + \frac{(1-\gamma)}{\gamma} CTe\alpha - \frac{(1-\gamma)VC}{\gamma} \ln \frac{B\alpha}{C} \tag{12}$$

$$G_{TB}^* - \mathbb{E}\left[G_{\text{TV.EXP3.M}}\right] \leq (e-1)\gamma G_{TB}^* + \frac{(1-\gamma)}{\gamma} CTe\alpha + \frac{(1-\gamma)VC}{\gamma} \ln \frac{C}{B\alpha}$$

$$\leq (e-1)\gamma TB + \frac{(1-\gamma)}{\gamma} CTe\alpha + \frac{VC}{\gamma} \ln \frac{C}{B\alpha} - VC \ln \frac{C}{B\alpha} \qquad \text{by } G_{TB}^* \leq TB$$

$$\leq (e-1)\gamma TB + \frac{1}{\gamma} CTe\alpha + \frac{VC}{\gamma} \ln \frac{C}{B\alpha}. \tag{13}$$

In Eqn. (12), we use the fact that $\sum_{v=1}^{V} \sum_{t=T_v}^{T_{v+1}-1} \sum_{c=1}^{C} g_t(c) \leq \frac{C}{B} G_{TB}^* = \frac{C}{B} \sum_{v=1}^{V} \sum_{t=T_v}^{T_{v+1}-1} \sum_{c \in A^*} g_t(c)$.

**Algorithm 5** TV.EXP3.M algorithm

---

Input: $\gamma = \sqrt{\frac{C\ln(C/B)}{(e-1)BT}}, \alpha = \frac{1}{T}, C$ #categorical choice, $T$ #max iteration, $B$ #multiple play

1: Init $\omega_c = 1, \forall c = 1...C$ and denote $\eta = (\frac{1}{B} - \frac{\gamma}{C})\frac{1}{1-\gamma}$
2: **for** $t = 1$ to $T$ **do**
3:    **if** $\arg\max_{c \in [C]} \omega_c \geq \eta \sum_{c=1}^{C} \omega(c)$ **then**
4:       $\nu$ s.t. $\frac{\nu}{\eta} = \sum_{\omega_t(c) \geq \nu} \nu + \sum_{\omega_t(c) < \omega_t(c)} \omega_t(c)$
5:       Set $S_0 = (c : \omega_t(c) \geq \nu)$ and $\omega_t(S_0) = \nu$
6:    **else**
7:       Set $S_0 = \emptyset$
8:    **end if**
9:    Compute $p_t^c = B\left((1-\gamma)\frac{\omega_c}{\sum_{c=1}^{C}\omega_c} + \frac{\gamma}{C}\right), \forall c$
10:   A batch of $B$ categorical choices $S_t = [c_t^1, c_t^2, ..., c_t^B] = \text{DepRound}\left(B, [p_t^1 p_t^2 .... p_t^C]\right)$
11:   Observe the reward $g_t(c) = f(c_t = c)$ for $c \in S_t$
12:   $\hat{g}_t(c) = \frac{g_t(c)}{p_t^c}, \forall c \in S_t$ and $\hat{g}_t(c) = 0$ otherwise
13:   $\forall c \notin S_0$ : update $\omega_c = \omega_c \times \exp\left(B\gamma\hat{g}_t(c)/C\right) + \frac{e\alpha}{C}\sum_{i=1}^{C}\omega_c,$
14:   $\forall c = S_0$ : update $\omega_c = \omega_c + \frac{e\alpha}{C}\sum_{i=1}^{C}\omega_c$
15: **end for**

---

Set $\gamma = \min\left\{1, \sqrt{\frac{C\ln(C/B)}{(e-1)BT}}\right\}$ and $\alpha = \frac{1}{T}$. Then, we have the cumulative regret over all iterations and considering the best element in a batch

$$\mathbb{E}[R_{TB}] = \mathbb{E}\left[\sum_{v=1}^{V}\sum_{t=T_v}^{T_{v+1}-1}\overbrace{\max_{\forall c \leq C} g_t(c)}^{\text{optimal arms}}\right] - \mathbb{E}\left[\sum_{v=1}^{V}\sum_{t=T_v}^{T_{v+1}-1}\overbrace{\max_{\forall c \in S_t} g_t(c)}^{\text{best arm in a batch } S_t}\right] \tag{14}$$

$$\leq \frac{1}{B}G_{TB}^* - \frac{1}{B}\mathbb{E}[G_{\text{TV.EXP3.M}}] \tag{15}$$

$$\leq \frac{1}{B}\sqrt{\frac{C(e-1)\ln(C/B)}{BT}}T + \frac{1}{B}\frac{\sqrt{C(e-1)BT}}{\sqrt{\ln(C/B)}}e + \frac{1}{B}\frac{VC}{\sqrt{\frac{C\ln(C/B)}{(e-1)BT}}}\ln\frac{CT}{B} \tag{16}$$

$$\leq [1 + e + V]\sqrt{(e-1)\frac{CT}{B}\ln\frac{CT}{B}} \tag{17}$$

where we obtain Eqn. (15) because the best gain should be greater than the average gain of a batch $\sum_{v=1}^{V}\sum_{t=T_v}^{T_{v+1}-1}\max_{\forall c \in S_t} g_t(c) \geq \frac{1}{B}\sum_{v=1}^{V}\sum_{t=T_v}^{T_{v+1}-1}\sum_{c \in S_t} g_t(c)$.

Given the number of changing points in our reward function is bounded $V \ll \sqrt{T}$ and the number of category $C$ is a constant, our regret bound achieves sublinear regret rate with the number of iterations $T$, i.e., $\lim_{T\to\infty}\frac{\mathbb{E}[R_{TB}]}{T} = 0$.

$\square$

In the above derivation, $T$ refers to the number of bandit updates which is equivalent to the number of $t_{\text{ready}}$ in PB2 setting [52].

## D.2 Theoretical Derivations for PB2-Mult

We adapt Lemma 1 in [52] to handle categorical variables. We first restate some notations used in [52]. Let $F_t^{c_t}(x_t)$ be an objective function under a given set of continuous hyperparameters $x_t$ and categorical variable $c_t$ at timestep $t$. An example of $F_t^{c_t}(x_t)$ could be the reward for a deep RL agent. When training for a total of $T$ steps, our goal is to maximize the final performance $F_T^{c_T}(x_T)$. We formulate this problem as optimizing the time-varying black-box reward function $f_t$, over $\mathcal{D}$.

**Lemma 4.** *Maximizing the final performance $F_T$ of a model with respect to a given continuous hyperparameter schedule $\{x_t\}_{t=1}^T$ and a categorical hyperparameter schedule $\{c_t\}_{t=1}^T$ is equivalent to maximizing the time-varying black-box function $f_t(x_t)$ and minimizing the corresponding cumulative regret $r_t(x_t)$,*

$$\max F_T^{c_T}(x_T) = \max \sum_{t=1}^{T} f_{c_t}(x_t) = \min \sum_{t=1}^{T} r_t(x_t). \tag{18}$$

*Proof.* At each $t_{\text{ready}}$, we select a categorical variable $c_t \in \{1, ..., C\}$ and a continuous variable $x_t \in \mathcal{D}^{c_t}$. We would emphasize that the continuous variables are conditioned on the choice of $c_t$. We observe and record noisy observations, $y_t = f_{c_t}(x_t) + \epsilon_t$, where $\epsilon_t \sim \mathcal{N}(0, \sigma^2 \mathbf{I})$ for some fixed $\sigma^2$. The function $f_t$ represents the change in $F_t$ after training for $t_{\text{ready}}$ steps, i.e. $f_{c_t}(x_t) = F_t^{c_t}(x_t) - F_{t-t_{\text{ready}}}^{c_{t-t_{\text{ready}}}}(x_{t-t_{\text{ready}}})$. We define the best choice at each timestep as $x_t^* = \arg\max_{x_t \in \mathcal{D}^{c_t}} f_{c_t}(x_t)$, and $c_t^* = \arg\max_{c_t, x_t} f_{c_t}(x_t)$. The intermediate regret of each decision is defined as $r_t = f_{c_t^*}(x_t^*) - f_{c_t}(x_t)$ where $f_{c_t^*}(x_t^*)$ is an unknown constant.

We have a reward at the starting iteration $F_1^{c_1}(x_1)$ as a constant that allows us to write the objective function as:

$$F_T^{c_T}(x_T) - F_1^{c_1}(x_1) = \underbrace{F_T^{c_T}(x_T) - F_{T-1}^{c_{T-1}}(x_{T-1})}_{f_{c_{T-1}}(x_{T-1})} + ... + \underbrace{F_3^{c_3}(x_3) - F_2^{c_2}(x_2)}_{f_{c_2}(x_2)} + \underbrace{F_2^{c_2}(x_2) - F_1^{c_1}(x_1)}_{f_{c_1}(x_1)}. \tag{19}$$

Therefore, maximizing the left of Eqn. (19) is equivalent to minimizing the cummulative regret as follows:

$$\max[F_T^{c_T}(x_T) - F_1^{c_1}(x_1)] = \max \sum_{t=1}^{T} F_t^{c_t}(x_t) - F_{t-1}^{c_{t-1}}(x_{t-1}) = \max \sum_{t=1}^{T-1} f_{c_t}(x_t) = \min \sum_{t=1}^{T-1} r_t(x_t)$$

where we define $f_{c_t}(x_t) = F_t^{c_t}(x_t) - F_{t-1}^{c_{t-1}}(x_{t-1})$, the regret $r_t = f_{c_t^*}(x_t^*) - f_{c_t}(x_t)$. $\qquad \square$

We then restate Theorem 2 in [52] which will be used in deriving the convergence guarantee for PB2-Mult.

**Theorem 5.** *(Theorem 2 in [52]) Let the domain $\mathcal{D} \subset [0, r]^d$ be compact and convex where $d$ is the dimension and suppose that the kernel is such that $f_t \sim GP(0, k)$ is almost surely continuously differentiable and satisfies Lipschitz assumptions $\forall L_t \geq 0, t \leq \mathcal{T}, \forall j \leq d, p(\sup \left| \frac{\partial f_t(\mathbf{x})}{\partial \mathbf{x}^{(j)}} \right| \geq L_t) \leq ae^{-(L_t/b)^2}$ for some $a, b$. Pick $\delta \in (0, 1)$, set $\beta_T = 2\log\frac{\pi^2 T^2}{2\delta} + 2d\log rdbT^2 \sqrt{\log\frac{da\pi^2 T^2}{2\delta}}$ and define $C_1 = 32/\log(1 + \sigma_f^2)$, the PB2 algorithm satisfies the following regret bound after $T$ time steps over $B$ parallel agents with probability at least $1 - \delta$:*

$$R_{TB} = \sum_{t=1}^{T} f_t(\mathbf{x}_t^*) - \max_{b \leq B} f_t(\mathbf{x}_{t,b}) \leq \sqrt{C_1 T \beta_T \left(\frac{T}{\tilde{N}B} + 1\right) \left(\gamma_{\tilde{N}B} + \left[\tilde{N}B\right]^3 \omega\right)} + 2$$

*the bound holds for any block length $\tilde{N} \in \{1, ..., T\}$ and $B \ll T$.*

Next, we are going to derive the main theorem of the paper.

**Theorem 6.** *(Theorem 2 in the main paper) Let the domain for continuous variables $\mathcal{D} \subset [0, r]^d$ be compact and convex where $d$ is the dimension and suppose that the kernel is such that $f_t \sim GP(0, k)$ is almost surely continuously differentiable and satisfies Lipschitz assumptions $\forall L_t \geq 0, t \leq \mathcal{T}, \forall j \leq d, p(\sup \left| \frac{\partial f_t(\mathbf{x})}{\partial \mathbf{x}^{(j)}} \right| \geq L_t) \leq ae^{-(L_t/b)^2}$ for some $a, b$.*

*Assume the reward distributions to change at arbitrary time instants, but the total number of change points is no more than $V$. Set $\alpha = \frac{1}{T}$, $\gamma = \min\left\{1, \sqrt{\frac{C\ln(C/B)}{(e-1)BT}}\right\}$, $\beta_T = 2\log\frac{\pi^2 T^2}{2\delta} + 2d\log rdbT^2 \sqrt{\log\frac{da\pi^2 T^2}{2\delta}}$, pick $\delta \in (0, 1)$ and define $C_1 = 32/\log(1 + \sigma_f^2)$, the PB2-Mult algorithm satisfies the following regret bound after $T$ time steps over $B$ parallel agents with probability*

*at least* $1 - \delta$:

$$\mathbb{E}[R_{TB}] \le [1 + e + V] \sqrt{(e-1)\frac{CT}{B} \ln \frac{CT}{B}} + \sqrt{C_1 T_{c_t^*} \beta_{T_{c_t^*}} \left(\frac{T_{c_t^*}}{\tilde{N}B} + 1\right) \left(\gamma_{\tilde{N}B} + [\tilde{N}B]^3 \omega\right)} + 2.$$

*The bound holds for any* $\tilde{N} \in \{1, ..., T_{c_t^*}\}$ *and* $V \ll \sqrt{T}$.

*Proof.* We expand the cumulative regret and optimize it using time-varying GP bandit optimization [7, 52]

$$R_{TB} = \sum_{t=1}^{T} f^* - \max_{b=1...B} \sum_{t=1}^{T} f_{c_{t,b}}(x_{t,b})$$

$$= \max_{b=1,...,B} \sum_{t=1}^{T} f_{c_{t,b}^*}(x_{t,b}) - \max_{b=1,...,B} \sum_{t=1}^{T} f_{c_{t,b}}(x_{t,b}) + \sum_{t=1}^{T} f^* - \max_{b=1,...,B} \sum_{t=1}^{T} f_{c_{t,b}^*}(x_{t,b})$$

where $b \in \{1, ..., B\}$ is an agent's index being trained in parallel, $f_{c_t^*}(x_t) = \max_{\forall c_t \in \{1,...,C\}} f_{c_t}(x_t)$ and $c_t^* = \arg\max_{\forall c \in \{1,...C\}} f_c(x_t)$ is the optimal categorical choice at iteration $t$.

We bound the two terms separately as follows. The first term is bounded by Theorem 1. We assume the process of generating the arm $c_t$'s reward $f_{c_t}(.) := f_{c_t}(x_t)$ is by the "adversary" that TV.EXP3.M does not have the direct control on the selection of $x_t$. Particularly, $x_t$ will be chosen by TV-GP-BUCB (as part of PB2) in Eqn. (3). We take the expectation of the first term to have

$$\mathbb{E}\left[\max_{b=1...B} \sum_{t=1}^{T} \underbrace{f_{c_{t,b}^*}(.)}_{\text{pull optimal arm}}\right] - \mathbb{E}\left[\max_{b=1...B} \sum_{t=1}^{T} \underbrace{f_{c_{t,b}}(.)}_{\text{pull arm } c_t}\right] = \mathbb{E}\left[\sum_{t=1}^{T} \underbrace{f_{c_t^*}(.)}_{\text{pull optimal arm}}\right] - \mathbb{E}\left[\max_{b=1...B} \sum_{t=1}^{T} \underbrace{f_{c_{t,b}}(.)}_{\text{pull arm } c_t}\right]$$

$$= \mathbb{E}\left[\tilde{R}_{TB}\right]$$

$$\le [1 + e + V]\sqrt{(e-1)\frac{CT}{B} \ln \frac{CT}{B}}$$

where $R_{TB}$ is cumulative regret of the TV.EXP3.M, defined in Theorem 1.

Assuming the best arm (the best categorical choice) $c_t^*$ can be identified by TV.EXP3.M in Theorem 1. The second term is the regret bound presented in Theorem 2 of the PB2 [52]:

$$\sum_{t=1}^{T} f_t^* - \max_{b=1...B} \sum_{t=1}^{T} f_{c_{t,b}^*}(x_{t,b}) = \sum_{t=1}^{T} f_t^* - \max_{b=1...B} \sum_{t=1}^{T} f_{c_t^*}(x_{t,b})$$

$$\le \underbrace{\sqrt{C_1 T_{c_t^*} \beta_{T_{c_t^*}} \left(\frac{T_{c_t^*}}{\tilde{N}B} + 1\right) \left(\gamma_{\tilde{N}B} + [\tilde{N}B]^3 \omega\right)} + 2}_{\mathcal{O}(PB2)}$$

where $T_{c_t^*}$ denotes the number of times the optimal category $c_t^*$ (or optimal arm) is selected. We follow [7, 52] to denote a block length $\tilde{N} \in \{1, ....T_{c_t^*}\}$ in which the function does not change significantly. In the above equation, $\omega \in [0, 1]$ is a time-varying hyperparameter which is estimated by maximizing the GP log marginal likelihood, $B$ is a batch size or the number of population agents, $\gamma_{\tilde{N}B}$ is the maximum information gain [74] defined within a block $\tilde{N}$ over $B$ parallel agents.

Note that the theoretical result for PB2 comes with the additional smoothness assumption on the kernel $k$ that holds for some $(a, b)$ and $\forall L_t \ge 0$. The kernel satisfies for all dimensions $j = 1, ..., d$

$$\forall L_t \ge 0, t \le \mathcal{T}, p(\sup \left|\frac{\partial f_t(\mathbf{x})}{\partial \mathbf{x}^{(j)}}\right| \ge L_t) \le a e^{-(L_t/b)^2}. \tag{20}$$

We combine the two terms to obtain the final regret bound of the PB2-Mult algorithm:

$$\mathbb{E}[R_{TB}] \le [1 + e + V]\sqrt{(e-1)\frac{CT}{B} \ln \frac{CT}{B}} + \sqrt{C_1 T_{c_t^*} \beta_{T_{c_t^*}} \left(\frac{T_{c_t^*}}{\tilde{N}B} + 1\right) \left(\gamma_{\tilde{N}B} + [\tilde{N}B]^3 \omega\right)} + 2.$$

The final regret bound is a summation of two sub-linear terms. Therefore, the regret bound grows sublinearly with the number of iterations $T$, i.e., $\lim_{T \to \infty} \frac{\mathbb{E}[R_{TB}]}{T} = 0$. This is under a common assumption [7, 52] that the time-varying function is correlated, i.e., we have $\omega \to 0$, thus $\tilde{N} \to T$. We also note that when $T \to \infty$, then $T_{c_T^*} \to \infty$ due to Theorem 1. $\square$

### D.3  Gradients

We optimize the GP hyperparameters by maximizing the log marginal likelihood [58]. We fit the GP hyperparameters by maximizing their posterior probability (MAP), $p(\sigma_x, \sigma_t \mid \mathbf{X}, \mathbf{t}, \mathbf{y}) \propto p(\sigma_x, \sigma_t, \mathbf{X}, \mathbf{t}, \mathbf{y})$, which, thanks to the Gaussian likelihood, is available in closed form

$$\mathcal{L} := \ln p(\mathbf{y}, \mathbf{X}, \mathbf{t}, \theta) = \frac{1}{2} \mathbf{y}^T \left( \mathbf{K} + \sigma_y^2 \mathbf{I}_N \right)^{-1} \mathbf{y} - \frac{1}{2} \ln \left| \mathbf{K} + \sigma_y^2 \mathbf{I}_N \right| + \ln p_{hyp}(\theta) + \text{const} \quad (21)$$

where $\mathbf{I}_N$ is the identity matrix in dimension $N$ (the number of points in the training set), and $p_{hyp}(\theta)$ is the prior over hyperparameters. We optimize Eqn. (21) with a gradient-based optimizer, providing the analytical gradient to the algorithm.

Our time-varying CoCaBO kernel is defined as

$$k_z(z, z') = (1 - \lambda)(k_{xt} + k_{ct}) + \lambda k_{xt} k_{ct}$$

where $k_{xt} = k_{\text{Continuous}}(x, x') \times k_{\text{time1}}(t, t')$, $k_{ct} = k_{\text{Categorical}}(c, c') \times k_{\text{time2}}(t, t')$, $k_{\text{Continuous}}(x, x') = \sigma_1 \times \exp\left(-\frac{||x - x'||^2}{l}\right)$, $k_{\text{Categorical}}(c, c') = \frac{\sigma_2}{C} \sum \mathbb{I}(c, c')$, $k_{\text{time1}}(t, t') = (1 - \epsilon_1)^{\frac{|t-t'|}{2}}$ and $k_{\text{time2}}(t, t') = (1 - \epsilon_2)^{\frac{|t-t'|}{2}}$.

Our hyperparameters include $\theta = \{\epsilon_1, \epsilon_2, l, \sigma_1, \sigma_2, \lambda\}$. We need to compute the gradients of $\frac{\partial \mathcal{L}}{\partial \epsilon_1}, \frac{\partial \mathcal{L}}{\partial \epsilon_2}, \frac{\partial \mathcal{L}}{\partial l}, \frac{\partial \mathcal{L}}{\partial \sigma_1}, \frac{\partial \mathcal{L}}{\partial \sigma_2}, \frac{\partial \mathcal{L}}{\partial \lambda}$ as follows:

- The gradient of $\frac{\partial \mathcal{L}}{\partial \epsilon_1}$

$$\frac{\partial \mathcal{L}}{\partial \epsilon_1} = \frac{\partial \mathcal{L}}{\partial k_z} \times \frac{\partial k_z}{\partial \epsilon_1}$$

$$\frac{\partial k_z}{\partial \epsilon_1} = (1 - \lambda) k_{\text{Continuous}}(x, x') \frac{\partial k_{\text{time1}}}{\partial \epsilon_1} + \lambda k_{ct} k_{\text{Continuous}}(x, x') \frac{\partial k_{\text{time1}}}{\partial \epsilon_1}$$

$$\frac{\partial k_{\text{time1}}}{\partial \epsilon_1} = -\frac{|t - t'|}{2} (1 - \epsilon_1)^{\frac{|t-t'|}{2} - 1}$$

- The gradient of $\frac{\partial \mathcal{L}}{\partial \epsilon_2}$

$$\frac{\partial \mathcal{L}}{\partial \epsilon_2} = \frac{\partial \mathcal{L}}{\partial k_z} \times \frac{\partial k_z}{\partial \epsilon_2}$$

$$\frac{\partial k_z}{\partial \epsilon_2} = (1 - \lambda) k_{\text{Categorical}}(c, c') \frac{\partial k_{\text{time2}}}{\partial \epsilon_2} + \lambda k_{xt} k_{\text{Categorical}}(c, c') \frac{\partial k_{\text{time2}}}{\partial \epsilon_2}$$

$$\frac{\partial k_{\text{time2}}}{\partial \epsilon_2} = -\frac{|t - t'|}{2} (1 - \epsilon_2)^{\frac{|t-t'|}{2} - 1}$$

- The gradient of $\frac{\partial \mathcal{L}}{\partial l}$

$$\frac{\partial \mathcal{L}}{\partial l} = \frac{\partial \mathcal{L}}{\partial k_z} \times \frac{\partial k_z}{\partial l}$$

$$\frac{\partial k_z}{\partial l} = (1 - \lambda) k_{\text{time1}}(t, t') \frac{\partial k_{\text{Continuous}}}{\partial l} + \lambda k_{ct} k_{\text{time1}}(t, t') \frac{\partial k_{\text{Continuous}}}{\partial l}$$

$$\frac{\partial k_{\text{Continuous}}}{\partial l} = \frac{||x - x'||^2}{l^2} k_{\text{Continuous}}(x, x')$$

- The gradient of $\frac{\partial \mathcal{L}}{\partial \sigma_1}$

$$\frac{\partial \mathcal{L}}{\partial \sigma_1} = \frac{\partial \mathcal{L}}{\partial k_z} \times \frac{\partial k_z}{\partial \sigma_1}$$

$$\frac{\partial k_z}{\partial \sigma_1} = (1 - \lambda)\, k_{\text{time1}}(t, t') \frac{\partial k_{\text{Continuous}}}{\partial \sigma_1} + \lambda k_{ct} k_{\text{time1}}(t, t') \frac{\partial k_{\text{Continuous}}}{\partial \sigma_1}$$

$$\frac{\partial k_{\text{Continuous}}}{\partial \sigma_1} = k_{\text{Continuous}}(x, x')$$

- The gradient of $\frac{\partial \mathcal{L}}{\partial \sigma_2}$

$$\frac{\partial \mathcal{L}}{\partial \sigma_2} = \frac{\partial \mathcal{L}}{\partial k_z} \times \frac{\partial k_z}{\partial \sigma_2}$$

$$\frac{\partial k_z}{\partial \sigma_2} = (1 - \lambda)\, k_{\text{time2}}(t, t') \frac{\partial k_{\text{Categorical}}}{\partial \sigma_2} + \lambda k_{xt} k_{\text{time2}}(t, t') \frac{\partial k_{\text{Categorical}}}{\partial \sigma_2}$$

$$\frac{\partial k_{\text{Categorical}}}{\partial \sigma_2} = k_{\text{Categorical}}(c, c')$$

- The gradient of $\frac{\partial \mathcal{L}}{\partial \lambda}$

$$\frac{\partial \mathcal{L}}{\partial \lambda} = \frac{\partial \mathcal{L}}{\partial k_z} \times \frac{\partial k_z}{\partial \lambda}$$

$$\frac{\partial k_z}{\partial \lambda} = -(k_{ct} + k_{xt}) + k_{ct} k_{xt}$$