# OpenReview forum: "Tuning Mixed Input Hyperparameters on the Fly for Efficient Population Based AutoRL"
_NeurIPS.cc/2021/Conference — NeurIPS 2021 Poster_

### Official Review · Reviewer_GNeX · 2021-07-11

**Rating:** 7
**Confidence:** 4

**Summary:**

The paper extends PB2 to allow for categorical hyperparameters in two different ways (direct integration into the kernel or two-step exploration depending on categorical choices). The authors show regret and convergence properties similar to PB2. In their evaluations, they show the benefits of integrating categorical hyperparameters by tuning the data augmentation alongside agent hyperparameters.

**Limitations And Societal Impact:**

Yes, adequately addressed in view of that the paper deals with a meta-problem and not with a specific application.

**Main Review:**

This is a well-written and technically sound paper extending an existing promising approach in hyperparameter optimization in RL. Incorporating categorical hyperparameters into PB2 in a way that allows for optimization with regards to dependencies between categorical and continuous variables will increase the utility of the algorithm and be of interest to its users. The paper gives the core insights into the extensions, mostly through its experiments which focus on the ProcGen benchmark. On one hand, this shows the improvements in a very challenging setting within RL. On the other, its computational expense leads to a less thorough exploration of the research questions. Overall an interesting addition.

## Novelty
The paper extends an existing approach in an important and useful way.

## Significance / Impact
PB methods are an important approach for hyperparameter optimization in AutoRL, so this will be very useful and probably have a large impact for existing users of automated HPO methods in RL (especially since it will also be integrated into Ray). The impact on the research community overall depends on how well AutoRL methods in general will be adopted.

I wonder a bit about the limited scope to AutoRL. PB2 and this extension could also be very well applied to other continuous adaptation problems, right? I think it would make the paper stronger and increase its impact outside of RL, if the authors could show the generic applicability to many problems and not only AutoRL.

## Soundness (method and experimental setup)
The authors use a very thorough setup for the toy example, but time constraints of course limit this on ProcGen. Still, 5 repetitions are an acceptable compromise and including the PB2 results without reruns is a fine trade-off in this case.

The research questions are interesting, relevant to the method and answered well for the most part. I’m only a bit disappointed in the scaling study since it does not show how well it would perform with different population sizes. This could have been done on a different benchmark than ProcGen if time constraints were the deciding factor, maybe by reducing the number of ProcGen games used by one or two. That would have given the research questions beside improved performance more space.

As the paper extends PB2, it addresses points like the regret bounds of that paper as it should. Overall, the experiments are done soundly and answer the most important questions of this extension. A comparison to similar non-population-based methods (especially CoCaBO) would have been nice, but given the computational budget it is understandable the authors chose not to include any.

## Clarity
The authors used a clear language; critical points are addressed straightforwardly (e.g. details of the experiments) and the visuals help with understanding. Background is sufficiently explained to researchers familiar with RL methods. Overall very good.

## Minor Comments

* For Table 1, I wondered whether it is important in practice whether it is population-based or not. I think this is not as important as whether the methods are efficient on categorical and continuous spaces. If the authors think there is a relevant difference, the text should reflect that and explicitly mention why e.g. CaCoBO suffers from not being population-based.
* Similarly, in the discussion it is stated that the paper makes a strong case for population-based methods specifically. I don’t really see how this is true as there is no comparison to other HPO methods or a strong demonstration that dynamic hyperparameter settings are important on ProcGen. In fact, UCB-DrAC is very close to the PB2 extensions in performance. So I would argue the results are more indicative of the extensions improving over vanilla PB2 and PBT-style training being a viable option on ProcGen rather than making a case for it over any others.
* Sometimes the authors call s_t a state and sometimes an observation. Should be consistent.
* I wouldn’t call reward scaling an implementation detail as it can decide whether an agent can learn efficiently or not (Section 3.2)
* What is meant by “combinatorially large search space”? If the space includes continuous hyperparameters, it is infinitely large anyway.
* I think that it is very unfortunate to call the compact, convex subset $\mathcal{D}$ which is typically used for datasets. This confused me at first.
* Since $r$ is normally used for the reward in RL, I would propose to use $\rho$ for the regret to not confuse readers.

## Questions for Rebuttal

* Eq 3: Is it possible to also use other acquisition functions (such as EI) instead of LCB?
* Section 6.1: The synthetic benchmark was not quite clear to me since it seems that h_t^b does not really depend on t and therefore, there is no real advantage to use PBT/PB2 over other HPO approaches
* Section 6.2. Why have you chosen a linear regression model? Is it a reasonable assumption that a linear model can fit the data well?

**Time Spent Reviewing:**

4

---

> ### Author Response · Authors · 2021-08-06
> **Thank you for the review!**
>
> Thank you for the overall positive review, there are many encouraging comments here which we appreciate. We agree with most of the comments, and only ask what we need to do for the score to be an accept? When reading this, it seems the reviewer believes our work makes a meaningful contribution. We hope our responses to the minor concerns provide the necessary reassurance required.
>
> **Population-based methods**
>
> We realize this was not clear enough, so thank you for highlighting. PBT as a general framework has been shown to be particularly effective in RL problems, used in many high impact projects. For instance, DeepMind’s recent works on football (or soccer) from pixels and open ended learning [1,2], and the work using PBT to optimize MBRL, breaking the MuJoCo simulator [3] (we are not authors or affiliated with these, but noted the use of PBT in all cases).
>
> PBT style methods benefit from adaptive schedules, which are particularly useful in RL, since the data distribution is constantly changing as the agent learns and explores different regions of the state space. PBT style methods also benefit from exploiting in the weight space, which alone provides performance improvements which cannot be achieved by static hyperparameter search (see Jaderberg et al 2017). We would not expect PBT (and PB2-Rand, PB2-Mix etc) to necessarily outperform other hyperparameter optimization algorithms on non-RL tasks. Also PB2-Rand was not rigorously tested in this setting. Thus, we feel it makes sense to continue to improve PBT methods for RL since that is the area where they may have the highest impact.
>
> **Other acquisition functions**
>
> This is definitely the case. We could consider any BO acquisition function in the PB2 framework. The reason for using GP-UCB methods is to provide theoretical guarantees. The original PBT is largely a heuristic based method so this is a clear departure. However, there may be other methods, such as Thompson Sampling which could also be effective here. As far as we are aware, our paper is only the second work exploring this class of methods (after PB2-Rand), and we suspect that with enough attention the community could significantly improve upon our work with novel acquisition functions (amongst other areas). Also, the modelling of time could likely be significantly improved. For now, we are addressing a key limitation of PB2 in that it doesn’t consider categorical variables at all. Once this work has been completed, we will consider improving the method further, and we believe others in the community will too.
>
> **The toy example**
>
> The reviewer is correct, the toy example does not analyze the time-varying component and thus it could be solved by other hyperparameter optimization methods. The key point in this study is it is a comparison of *different PBT methods*. We are seeking to only evaluate one degree of freedom in this experiment, for clarity purposes, since PB2-Rand is the only baseline we are trying to improve upon. We considered including additional toy experiments adding these components, but felt it would confuse readers. Indeed, we can see from Fig 4.b) (and the many plots in the Appendix) that TV.EXP3.M is able to adapt to the time-varying nature of data augmentation in RL. We see in this plot that cutout-color was one of the strongest categories in the first half of training, and was frequently picked, but it was third worst in the second half and not picked at all.
>
> **Linear regression model**
>
> We used the linear regression model to study causal relationships between the hyperparameters and subsequent improvement in policy return. Linear regression is the obvious choice for this, it is interpretable and used throughout applied statistical studies. We do not think it is the best model if we wanted to *predict* the return change, as is typically required in machine learning, but instead we wish to have interpretable coefficients with t-statistics to see if the variables are linearly related. In our algorithm we have a different emphasis, as we wish to predict the expected change in return, so we use a nonlinear model (a GP), but this is harder to interpret.
>
> We believe this is sufficient to show why the method works. It directly addresses the problem we are trying to address and marks a solid step forward for this class of methods.
>
> Thank you for your time!
>
> [1] From Motor Control to Team Play in Simulated Humanoid Football. Liu et al, 2021.
>
> [2] Open-Ended Learning Leads to Generally Capable Agents. Open-Ended Learning Team, 2021.
>
> [3] On the Importance of Hyperparameter Optimization for Model-based Reinforcement Learning. Zhang et al 2021.

---

> > ### Comment · Reviewer_GNeX · 2021-08-16
> > **Reply**
> >
> > Thanks for the thorough answer.
> > My thoughts on this:
> >
> > I would like to encourage you to also study PB2 on other applications besides RL. It's somehow fine for this paper, but it definitely would have more impact if the paper would show also convincing results on other applications.
> >
> > I'm still not convinced that the toy example is really a good one. I see why you chose to design it the way you did; but I believe that a first toy example should nevertheless reflect the how problem setting and not only a single aspect.
> >
> > I strongly push back on your argument for linear regression. There are good reasons why (nearly) no one from the ML community uses linear regression as the interpretable method to go. First of all, linear models have problems to explain if there correlation effects -- however it is not directly clear to me whether this applies to HPO. The biggest problem is that if the accuracy of the model is not sufficient, the explanations are meaningless. Please also note that also in the applied statistic studies there are strong assumption on applying a linear model for interpretation of unknown functions, incl. how the data was sampled.

---

> > > ### Author Response · Authors · 2021-08-17
> > > **Follow up**
> > >
> > > Thank you for coming back!
> > >
> > > For the other experiments, we will think about this. We most likely cannot get results before decisions are made - but - we plan to add our methods to the ray library once published. When we do this, we will likely create some minimal examples.
> > >
> > > Regarding the linear model, we think the reviewer maybe misunderstood here and we take full responsibility for that. The linear regression is *not part of our method* it is solely used *to quantify the relationship between hyperparameters and training performance at a very high level*, essentially just a correlation. We disagree with the claim no one ML uses this because in fact - as far as we are aware - no one uses anything at all in the hyperparameter optimization community! Indeed, almost all papers we have seen simply show the learned schedules (or static values) of the agents alongside the performance, and then make some comment along the lines of "our method used these hyperparameters and the performance was better so they must be good". Not going to provide specific references, but feel free to look at the papers we cited in related work.
> > >
> > > In our work we went a step further and *looked at the data to try and see if there was actually a relationship*. We understand there are more recent methods to deal with multicollinearity, and we could have used some advanced statistical techniques, but that is surely beyond the scope here. More complex models will also come with drawbacks, potentially hallucinating relationships where none exist without extreme care and attention, while also potentially over complicating our work which already contains significant technical content *in the actual method*. What we are doing here is providing some evidence that our method is doing something sensible, which the results seem to indicate.
> > >
> > > With this in mind, would it be sufficient to make it clearer that this method is limited, and thus only indicative not necessarily a guarantee of the relationships we show? We can also clarify that we never use these predictions for anything, it is just a statistic to demonstrate *some* evidence of dependence in Procgen, which had never previously been explored (by anyone, as far as we know).
> > >
> > > Please let us know if there is more we can clarify. If not, then given your initial review that "The paper extends an existing approach in an important and useful way", we would greatly appreciate it if it was now possible to consider providing us with an accept rating.
> > >
> > > Thank you!

---

> > > > ### Comment · Reviewer_GNeX · 2021-08-17
> > > > **References for explaining and quantifying HPO effects**
> > > >
> > > > > no one uses anything at all in the hyperparameter optimization community!
> > > >
> > > > Although it is not perfectly clear to me how to do it properly for dynamic approaches such as PB2, techniques for studying the relationship between hyperparameters and performance do exist in the HPO community, e.g.:
> > > >
> > > > * http://proceedings.mlr.press/v32/hutter14.html
> > > > * https://www.kdd.org/kdd2018/accepted-papers/view/hyperparameter-importance-across-datasets
> > > > * https://ojs.aaai.org/index.php/AAAI/article/view/10657
> > > > * https://openreview.net/forum?id=lZr9s1x0mE
> > > >
> > > > Nevertheless, with the agreed changes, I'm happy to increase my score to 7.

---

> > > > > ### Author Response · Authors · 2021-08-17
> > > > > **Thank you for the references**
> > > > >
> > > > > Thank you for sharing these references. We will read them and if possible borrow some ideas for analyzing hyperparameters in RL. Thus far, the only widespread studies we are aware of are those like [Andrychowicz et al](https://openreview.net/forum?id=nIAxjsniDzg) which study a wide variety of hyperparameters w.r.t final performance for PPO.  Their work made a significant contribution, and this was reflected in it being presented as an oral at ICLR 2021, but it does not tell us everything, especially when hyperparameters vary over time. They also do not consider the data augmentation setting, which is very new in RL but now used in a variety of state of the art methods (e.g. RAD, DrQ, SPR). They also do not consider statistical tests, just use the final performance of a static hyperparameter as a proxy.
> > > > >
> > > > > In addition, very few studies have considered the importance of hyperparameters for *generalization* in RL, the only work we are aware of is [this workshop paper](https://arxiv.org/abs/1906.00431). In our case we provided the analysis in support of a new method, but it is likely worthwhile to study this more rigorously in the future.
> > > > >
> > > > > We think there is a great deal of work to be done here!

---

### Official Review · Reviewer_QwWX · 2021-07-15

**Rating:** 6
**Confidence:** 2

**Summary:**

The paper presents a population-based approach for learning hyperparameter schedules on the fly. Its contribution to the field is to allow both continuous and categorical variables to be optimized in a non-random way.

**Limitations And Societal Impact:**

There's an acknowledgement of limitations (i.e. that the performance wasn't that great).

Societal Impact: N/A

**Main Review:**

### Overview

I had trouble understanding the paper, in part due to my background (no prior experience with PPO and PB2), but I’ve provided some suggestions that I believe would have helped with my understanding.

The justification for where this method fits among other population-based methods was clear and at a very high level the theory seemed like a great advancement, but most of the theory was in the appendix, which I didn’t review.

The empirical evidence was not great. I’m not convinced the performance was meaningfully different between the baseline and the two new methods. The theory might justify the poor performance, but I didn’t assess the theory in detail.

### Major

* The new approaches (PB2-Mult and PB2-Mix) were both within 1 standard deviation of the baseline (PB2-Rand) according to Table 2. The PB2-Mix improvement was statistically significant (line 298), but the p-value was a barely acceptable 0.044 (i.e. if PB2-Mix was equal to the baseline, we’d see a difference as large as this once every 23 times we ran the experiment). Furthermore, if a p-value is being used to consider significance, it is often scaled by the number of comparisons done (i.e. two: PB2-Mult and PB2-Mix), which would put this above the significance threshold.

### Clarity

* Lines 134–136 consider a *convex* hyperparameter space. Why is this assumption being made? This seems like a very strong assumption I’d like to see defended (even if just to say this is the assumption made by prior work, e.g. PBT).
* Figure 1 (near line 140) was not helpful to me. I'm not sure exactly how to improve it, but I think I would have appreciated seeing the entire pipeline of hyperparameter selection and evaluation. (Perhaps familiarity with the background, such as the Procgen protocol mentioned in line 286–287, would obviate this.)
* I didn’t find algorithm 2 (near line 198) to provide the right amount of high-level detail. I had to go to the appendix because on its own, Alg2 didn’t give me enough understanding. For example, I thought “normalize $\omega$” might be a simple [0,1] normalization, but it’s a more complex scaling that ensures no $\omega_c$ is larger than $\eta$ percent of the sum of weights. Perhaps indicate that $\eta$ is used as a scaling factor? Alg2 doesn’t indicate what $\eta$ is used for.
* I suggest wording line 198 as “Algorithm 5 *in* Sec. D.1” to indicate that Alg5 is in the appendix. I assumed Algorithm 5 was coming later in the paper after this high-level summary.

### Minor (typos and improved phrasing)

Some of these points are trivial, but you might find them helpful...
* Some acronyms were undefined. E.g. GP was used on lines 7, 42, 71, and 133 before finally being defined on line 158. DrAC on line 272 was never defined.
* Some hyphens were missing. Consider reading about how to [hyphenate compound adjectives](https://www.grammarly.com/blog/hyphen/). E.g. I suggest “population-based AutoRL”, “time-varying problem”, “high-level summary”, and “code-level hyperparameters”.
* Line 75 discusses data augmentation as a categorical variable. I’d have appreciated a couple of examples or a bit more background here.
* Line 100 says PPO is “one of the most widely used RL algorithms”. Compared to what? Sarsa? If “one of the most widely used” changes to “a widely used”, I can’t argue anymore and I won’t ask for a citation. :D
* Line 104 says “As can be seen,” but so far the paper has only pointed out one hyperparameter ($\epsilon$), so maybe delete that phrase.
* Line 106 would be more clear as “there is also often another hyperparameter, $\lambda$, in the advantage function [60].” (As written, it implied that $\lambda$ is the advantage function.)
* Lines 107–108: it’s unclear to me whether the “recent study” and the “recent work” referenced here are the same work or a different one. Since there’s only one citation [17], I think they both refer to the same study? Either the phrasing should be improved or this sentence needs another citation.
* On line 114, “they” should be deleted.
* Lines 114–115 refers to methods (plural), but only one citation [51] is provided.

**Time Spent Reviewing:**

5

---

> ### Author Response · Authors · 2021-08-06
> **Thank you for your feedback!**
>
> We thank you for your honesty, and despite our paper not being in an area of interest it was great to see you were still able to provide us with some great feedback, which is much appreciated.
>
> We do however disagree with some comments, in particular: “I’m not convinced the performance was meaningfully different between the baseline and the two new methods”. We address this in the first section below.
>
> **Statistical significance**
>
> Firstly in the synthetic task, there is clearly a large gain, with our new methods significantly outperforming PB2-Rand. No previous method has considered PBT with a time-varying bandit algorithm to select categories, and clearly it helps a lot in this simple setting. Furthermore, we include a link for reviewers to run the code and see that it is completely reproducible. This is not done very frequently in ML papers.
>
> For Procgen we agree (to an extent) that t-stats should be reduced based on the number of trials. Unfortunately our lab does not have significant compute resources, so we were not able to run this 23 times. Instead, we tried to strike a balance between showing a task the field is currently interested in, while also being able to run seven distinct environments. In almost all cases in ML (and RL in particular), researchers do a hyperparameter sweep before presenting statistical significance of their final method. In our case we only tested two variations, and we explicitly discussed the pros and cons of each in our discussion section. Of course, just because the standards are low elsewhere does not mean we should follow suit, and we should "be the change we want to see", but we think it is a little harsh to apply that criteria when it is just two different versions of a method, and we were open about the relative benefits of each in our discussion section. If we swept over all and picked based on the test performance somewhere in the Appendix, and then only presented the best versions in the main paper, then we absolutely agree with this.
>
> Alongside the quantitative results, we also provided intuition by showing the data does exhibit the type of dependence we are trying to model. We even show TV.EXP3.M is able to capture the time varying performance of different data augmentations Fig 4.b). Results can always be more emphatic, but we exhibit outperformance vs. our key baseline (PB2-Rand) and fix a problem with that method. While we also provided non-trivial theoretical results. To us, that seems like a worthy contribution to the conference.
>
> **Convex hyperparameter space assumption**
>
> We think this is a simple misunderstanding. In convex optimization, people consider convex functions over convex sets [Boyd et al 2004]. However, in our setting, *we don't assume convexity over the functions*. Instead, the convex set mentioned in line 134-136 refers to a standard setup, e.g., given any two points, it contains the whole line segment that joins them. Thus, this assumption avoids the fragmented and disjointed input space.
>
> In brief, our AutoRL settings considers the non-linear, non-convex, black-box optimizations [Srinivas et al 2010].
>
> **Minor points**
>
> These were all helpful, thank you.
> * Acronyms: noted!
> * Line 100 (PPO): Valid point! We will change this. Maybe we meant “deep RL”, but still, point taken.
> * Hyphens: we will certainly be bookmarking this link.
>
> Overall we hope that despite the relative lack of expertise, it is sufficiently clear to see our paper that addresses a clear problem with novel theoretical results and solid experiments. If this is the case, we hope the reviewer can at least provide us with a weak accept. If not, we would appreciate the opportunity to clarify additional areas of concern.
>
> Thank you!

---

> > ### Comment · Reviewer_QwWX · 2021-09-01
> > **Feedback reply**
> >
> > I've updated my score. I'm still not enthusiastic about the strength of the empirical results, but that's not the only focus of the paper, and the synthetic task at least shows a big difference.

---

### Official Review · Reviewer_3P6N · 2021-07-16

**Rating:** 6
**Confidence:** 4

**Summary:**

This paper extends the existing AutoRL algorithm, PB2, that was specifically designed to work with continuous hyperparameters by enabling it to efficiently search over both continuous and categorical hyperparameters of reinforcement learning solution methods. To do so, it first proposes a new multi-armed bandit algorithm called TV.EXP3.M (an extension of EXP3.M that varies with time) to select a subset of a given set of categories. Then, the paper incorporates this technique into the exploration stage of the PB2 algorithm to explore and select the categorical variables by taking into account the dependence of continuous variables on them. It takes two different approaches for modeling this dependency, one by incorporating it into the GP kernel and the other by filtering continuous variables based on the chosen categories that result in PB2-Mix and PB2-Mult algorithms. To evaluate the proposed algorithms, they first theoretically analyze PB2-Mult and TV.EXP3.M algorithm by showing they have cumulative regret bound and sublinear regret bound respectively. Then, they empirically evaluate their techniques along with PBT and original PB2 as the baselines on a toy environment and seven Procgen games and show that PB2-Mix achieves a similar or better performance compared to other baselines on these environments.

**Limitations And Societal Impact:**

The paper stated its limitations in section 6.

**Main Review:**

This is an interesting paper and the proposed contributions seem useful for the RL community. However, there are some issues with the work that preclude acceptance in the current form. To elaborate, (1) there are many missing details in the paper and weakly supported claims that make it hard to understand the paper and proposed techniques, and (2) the empirical study done in the paper fails to justify the stated contributions.

Detailed comments below:

There are some claims in the paper that needs further clarification. “As a result, many lauded approaches are impossible to reproduce without the vast resources required to sweep through hundreds or thousands of possible configurations [6].” The cited paper, [6], proposes an AutoML algorithm specifically designed to solve computer vision problems and not a reinforcement learning (RL) algorithm. This is happening while the main topic of the paragraph is about the practicality of RL techniques. This claim needs further justification.

The main claim of the paper is that the proposed algorithms enable the search over the space of categorical variables. Nevertheless, TV.EXP3.M seems to only consider selecting categorical variables that are binary. In another word, it can select whether a certain operation will be part of the algorithm or not. This observation is further supported by the hyperparameter search space that has been chosen for the categorical hyperparameters that are selecting a subset of different data augmentation operations. It is possible that I couldn’t understand this part really well, so I ask the following question. Is it possible to search over different data augmentations and optimizers using the proposed technique? If so, please describe how you would do it using the TV.EXP3.M algorithm. In this setting, you only get to select one optimizer out of n optimizers in your search space. But, it seems that TV.EXP3.M would select a subset of optimizers by taking multiple arms and this problem is not addressed in the paper.

In algorithms 2 and 5, it is not clear how the output of this algorithm, D_t, is chosen. It is not mentioned in the algorithms’ procedure, but it suddenly appears as output. Also, whereas this output is denoted by D_t in these algorithms, it is denoted by A_t in equation 4. Is there a reason for this notational difference here? Also, the style of “D” (D_{t-1}) used in equation 4 and the one in the output of Algorithm 4 (D_t) is different. Are they refer to different collections or datasets? Furthermore, there are not enough details about Algorithm 3 in the paper to understand how the filtering dataset procedure takes place. This part needs further explanation. “For the exploit step, the weights of the bottom agents are replaced by those from a randomly sampled agent from the set of best-performing agents, in what is called truncation selection.” The truncation selection procedure is not mentioned in the paper and there is not talk about its hyperparameters in the paper. Does this technique use any hyperparameter, such as the number of bottom agents that their weights should be replaced?


The experiments done in the paper do not fully support the stated contributions. In the abstract, it is mentioned: “where we show that explicitly modeling dependence between data augmentation and other hyperparameters improves generalization.” However, there is no baseline in the paper that uses TV.EXP3.M algorithm for selecting categories without modeling the dependence between continuous and categorical variables. In another part, it is stated “Despite using a small population size, we are able to train policies that match or outperform state-of-the-art methods which used a much larger grid search.” However, the evidence in the paper does not support this claim because the comparison with the UCB-DrAC algorithm in Table 2 is not justified. As is shown in Table 2, the proposed techniques in the paper and the UCB-DrAC algorithm have a highly similar performance to PB2-Mix and PB2-Mult, by their standard deviations overlapping each other in most of the environments. Also, based on the citation [21] in the paper, only 5 seeds for deep RL algorithms do not necessarily result in statistically significant results. So, to make a better comparison, the number of seeds needs to be increased.


Finally, this is a list of questions that need to be answered about the empirical study:
1. What is the value of \omega (in the equation on page 4) in your experiments? Also, how much your algorithm is sensitive to this hyperparameter?
2. A lot of new modules have been added to PB2 and eventually your algorithm to do AutoRL makes the original algorithm more computationally expensive. For instance, the computational cost of the gradient-based optimization approach that you take to optimize the GP hyperparameters can increase by the number of points in the training set. So, I wonder whether have you done any study to show how much computational cost these new modules add to the original algorithm that in this case is PPO?
3. Why this specific hyperparameter search space has been chosen for the empirical study? There are only four continuous hyperparameters in the paper, while there the used algorithm has much more hyperparameters to tune such as \beta_1 and \beta_2 for Adam optimizer. So, a better question would be: how does this algorithm scale as the number of hyperparameters increases?
4. What model is used in the paper to run the experiments and what specific hyperparameters have been used for training this model? The details of the neural architecture and Adam optimizer’s parameters are missing, such as the value of \beta_1 and the number of layers.


Minor comments:
1. Thus, if it wasn’t enough that PPO -> Thus, if it was not enough that PPO (Line 115)
2. Notation inconsistency in line 155: r_t is getting x_t as input while in line 153 it is defined without any input. So, replacing r_t to r_t(x_t) can clarify it here.
3. Line 90: “provides the agent with an observation s_t” -> “provides the agent with a state s_t”
4. Line 104: for a previous policy -> for the behavior policy. It is good to also add a brief definition of behavior policy such as the policy used to gather the data.
5. Line 259: PB2-R -> PB2-Rand
5. The name of the paper does not reflect the content of the paper and requires modification: “Tuning Mixed Input Hyperparameters on the Fly for Efficient Population Based AutoRL”. Based on the current name, it seems that the hyperparameter selection is designed for the AutoRL algorithm while it is an AutoRL algorithm that is designed for tuning the parameters of the RL algorithm.



**Time Spent Reviewing:**

15

---

> ### Author Response · Authors · 2021-08-06
> **Thank you very much!**
>
> Thank you for the detailed review and wide range of interesting comments. This review highlighted areas of our paper that absolutely needed to be clarified, so we appreciate the considerable time spent in finding them. We thank the reviewer in advance for reading our response, and hope that we have provided enough clarity that it is possible to provide us with a higher score as a consequence.
>
> **“claims in the paper that needs further clarification”**
>
> This is a good point and on reflection we agree. This comment was largely based on our experience in trying to reproduce RL results in other projects, which is actually what inspired us to look into PB2 style methods in the first place! However, we should be referring to specific published works rather than anecdotes, and thus should shift the emphasis. Instead, we can write that it is often challenging to reproduce results given the significant number of moving parts in RL algorithms. This is verified in [21] in our paper, where different implementations of algorithms lead to different performance (Fig. 6), as well as in [17].
>
> **“Is it possible to search over different data augmentations and optimizers using the proposed technique?”**
>
> Yes! The reviewer is correct that TV.EXP3.M does select a *batch* of arms, but, this batch corresponds to the agents not the values, *each agent only receives a single value*. So imagine we have a population size of M, then we select M arms. This is important because if we instead select 1 arm repeatedly M times, then there may be redundancy as we will *explore the same arms multiple times*. Regarding the specific question, we would have two categorical variables, and then select them both individually (independently) first, then condition on both when selecting continuous hyperparameters. If you see in CoCaBO (Ru et al 2020) this is a feature of the algorithm, it can handle “Multiple Continuous and Categorical Inputs”, but can also handle any combination (e.g. one category, one continuous like the toy example). PB2-Mix introduces a new way to use CoCaBO online, with a time-varying kernel and time-varying bandit. This means we can use it in PBT, which also benefits from weight copying during training.
>
> **Clarity of the technical content**
>
> These points are largely all typos on our part, which we appreciate the reviewer for highlighting:
> * *Where does D_t come from?* We follow the notations in PB2 (Parker-Holder et al 2020) to define D_t indicating all observed information up to iteration t. We will clarify this in Section 4.
> * *D_t vs. A_t in Eqn 4*:  D_t includes all continuous, categorical and the outcome y while A_t indicates the selection of categorical (or augmentation choices) at iteration t. For consistency, A_t should have been h_t, so this is a mistake on our part. We hope this did not cause too much confusion, and will certainly make this change.
> * *Style of D_t in Alg vs. Eqn 4*:  We thank the reviewer for pointing out this typo. This was unintentional and we will make the style consistent for D_t.
> * *Details of Alg3*: For the filtering, we say that we only use the data with the exact same categorical choice. For example, if the data was a pandas dataframe, df, we would do df[df[‘Cat’]==choice] and then fit the GP on that subset of the data. Thus, if we have C categories and each has been sampled equally, PB2-Mult uses 1/C of the available data. It is therefore more effective with a larger batch size w.r.t number of categories.
> * *Truncation selection details*: Great question, this should have been included in the paper. We used 25% for all settings, because we have a batch size that is a multiple of 4. So for most of our experiments we replace the bottom 1 agent with the top 1 agent. This number likely becomes a more important hyperparameter when we get to larger populations, but most PBT papers use something close to 25%.
>
> **Re: claim that modeling dependence between data augmentation and other hyperparameters improves generalization**
>
> We agree with the reviewer that this is not directly shown in the paper, more implied by the results. Indeed, we do actually have results for the baseline suggested, but did not include the ablation due to space constraints, and at the risk of confusing reviewers with a third new method. This may have been an oversight, but one that hopefully can be corrected in this discussion phase.
> For the toy problem (notebook link in the paper) this baseline is included. To select it, set cat_exp to be “exp3_indep”, this is using TV.EXP3.M but assuming independence. I appreciate you are busy, but if interested you can run this:
>
> ```
> args.search='PB2'
> args.cat_exp ='exp3_indep'
> results_pb2ind = get_results(args, N_trials)
> ```
> It should on average perform worse than cat_exp settings of ‘exp3_dep’ (PB2-Mult) and ‘cocabo’ (PB2-Mix). The setting in the notebook right now is a reduced experiment to facilitate speedy running, but if you increase the batch size/number of trials/max budget it is possible to run the experiments in the paper.
> As discussed, we do have this baseline for Procgen. See the test performance below, where "PB2-Indep" is the method proposed by the reviewer:
>
>
> | Environment | PB2-Rand | PB2-Indep  | PB2-Mix  |
> |---|---|---|---|
> | BigFish  | $8.2\pm0.1$ |  $9.9\pm2.8$ |  $10.6\pm1.9$ |
> | CaveFlyer | $5.8\pm1.8$ |  $5.2\pm0.6$ |  $6.0\pm0.7$ |
> | CoinRun | $8.1\pm0.1$ | $8.5\pm0.4$ | $8.1\pm0.2$  |
> | FruitBot  | $29.5\pm1.9$ | $27.0\pm1.5$  | $29.0\pm0.6$ |
> | Jumper  | $5.4\pm0.5$ | $7.1\pm1.2$  | $6.2\pm0.3$ |
> | Leaper  | $5.0\pm2.4$ | $5.1\pm0.4$  | $6.6\pm1.7$ |
> | StarPilot | $33.6\pm3.1$ | $37.8\pm3.8$  | $36.3\pm2.4$ |
> | PB2 Normalized | $100$ | $108.6$ | $112.1$ |
>
> As can be seen, TV.EXP3.M adds around 75\% of the improvement over PB2-Rand, but modelling the dependence adds an extra boost. This is another example of the strength of modelling the categorical variables which have previously been ignored by PBT style methods.
>
> **Claim to “match or outperform” existing methods that used grid search.**
>
> This claim is w.r.t Table 7 of Raileanu et al 2020 (the DrAC paper). Our results are comparable with UCB-DrAC as well as the best individual DrAC augmentation, and better than RAD, despite these results sweeping over several parameters. We can rephrase to “perform competitively” so as to *reduce the claim of outperforming*. Outperformance should not be necessary given we learn multiple hyperparameters from scratch, so we do not need to make that claim.
>
> Note however that our primary goal is to outperform PB2-Rand, and on an aggregate basis our results do show a statistically significant gain, alongside theoretical results.
>
> **Implementation Questions**
>
> 1. Value of \omega.
>
> This is set to: omega=1/(np.sum(numRounds)*10) in our code. This can be verified in the code attached.
>
> 2. Computational cost.
>
> We note the runtime can be improved in several ways (which we didn’t try), such as by fixing the kernel hyperparameters, using a sliding window of data, or using a faster GP implementation (an active area of research). This was not necessary for our experiments as the amount of time was a small fraction of the total training time. However, it may become important with large populations for longer training times.
>
> Indeed, there is inevitably a trade-off in terms of the desired efficiency of the explore step and the computational cost one is willing to incur. When the RL experiments are expensive to run, for example with vision based inputs, the explore steps (measured in a few seconds) make up a trivial amount of time compared to the several hours of training. However in the opposite setting where evaluations are extremely cheap, for example using a fast simulator with a large compute cluster, it may even be preferable to use the original PBT since there could be enough samples to explore the space with random search.
>
> We believe in most cases PB2-Mix would be the best approach, which is supported by our experiments. Furthermore, in cases where the dependence between continuous and categorical variables may be strong, PB2-Mult provides similar speed to the original PB2 and can be used effectively.
>
> **Specific hyperparameters chosen**
>
> The PPO parameters chosen are well-known to be important for learning, for example the clip parameter and learning rate influence the size of gradient updates, while the entropy influences the degree of exploration. These are typically tuned in RL papers, and have a large influence on performance (see [2] in our paper, Fig 10 for the PPO clip, Fig 81 for entropy and Fig 9 for learning rate). The DrAC coefficient was tuned in the DrAC paper, so we included that as well. We could have chosen others such as the discount factor and batch size, which may also be important. The Adam optimizer parameters have not been studied as extensively in the RL literature as I believe most take the defaults, however, with a larger budget it may be interesting to include them as well.
>
> **Optimizer and architecture details**
>
> In l.272 we state that we used the exact implementation from DrAC. However, we could have been clearer about this, and for completeness we will include details in the Appendix. The agent uses the IMPALA ResNet architecture commonly used in Procgen. We refer the reviewer to Raileanu et al (Appendix E) for additional details.
>
> **Additional comments**
>
> We appreciate the time taken to provide us with this additional feedback. We will certainly consider additional titles. If something better comes to mind during the discussion phase we will share it with all reviewers. We are open to suggestions if the reviewer has a strong opinion!
>
> Finally, we want to thank the reviewer again for their time both reviewing (15 hours) and in reading this response. If possible, please provide us with additional opportunities to clarify our paper. If we have already done so, then we hope it is possible to increase your score. Thank you!

---

> > ### Author Response · Authors · 2021-08-17
> > **Follow up**
> >
> > Hi Reviewer 3P6N,
> >
> > We understand reviewer load is higher than usual this year, and your initial 15 hour review was fantastic, so we greatly appreciate the additional time in reviewing our responses. We are writing in the hope to flag these responses once again.
> >
> > Reading your initial review, we got the impression that had it not been for clarification issues you would have been happy to accept our work. For example, the review said: "This is an interesting paper and the proposed contributions seem useful for the RL community". However, unfortunately, we made some mistakes in our presentation and omitted a useful baseline. In both cases these issues have been addressed:
> > * The many semantic issues have been clarified.
> > * We included an additional baseline for all Procgen games.
> >
> > As such, we were optimistic that you would be able to raise your score, possibly even to an accept. If this is not possible, please let us know so we have an opportunity to address the remaining concerns.
> >
> > Thank you!

---

> > > ### Comment · Reviewer_3P6N · 2021-09-02
> > > **Feedback**
> > >
> > > Thanks for the clarification of these points. Based on the current clarifications, I am going to increase my score to 6 if they also clarify why their results with only 5 runs are statistically significant, and why they made this decision. They clarified all of the other points, but clarification on this point is still missing.
> > >
> > >
> > > I am also still interested to see how this method works when we increase the number of categorical variables and consider other variables rather than augmentations that are used in the experiments. So, I encourage authors to add these results in the camera-ready version of the submission.

---

> > > > ### Author Response · Authors · 2021-09-02
> > > > **Thank you for your comment**
> > > >
> > > > Hi - Thank you for coming back.
> > > >
> > > > We agree that the deep RL matters paper raises a very alarming issue, that in some cases with five seeds it is possible to get statistically significant differences. However, our result is slightly different as the result we show is a statistical significance across *seven* games, where *each game* was run for five seeds. So we are showing significance in the normalized gain over PB2-Rand across *all 35 seeds*. With seven times more data it seems less likely that this is just an anomaly, but of course this does remain a possibility.
> > > >
> > > > We also refer the reviewer to a very new piece of work (not from any of the authors of our paper) which re-addresses many of these issues with deep RL performance analysis [here](https://arxiv.org/abs/2108.13264). The paper discusses the fact that for many large scale RL benchmarks it is simply not possible to run tens or hundreds of seeds, but instead it is possible to provide more robust statistics (rather than point estimates). They focus specifically on ProcGen in Section 5.3, and provide some new metrics for fairer comparison. We have run these new metrics using the open sourced colab and will include the results either in the Appendix or the main body of the paper. Based on this, PB2-Mix has a ~0.68 probability of improvement over PB2-Rand with a confidence interval of [0.51/0.84]. This seems to us like a reasonable improvement, compared to the results in the paper (Fig 12, right side).
> > > >
> > > > Regarding additional experiments, we would definitely like to include another benchmark/setting to increase the impact of our work. As mentioned elsewhere, we plan to open source the code as an update to the current Ray PB2 implementation, and will likely include smaller examples with that if possible.
> > > >
> > > > Thank you again for your time!

---

> > > > > ### Comment · Reviewer_3P6N · 2021-09-02
> > > > > **Improved my score**
> > > > >
> > > > > Thanks for introducing this paper. Based on this final comment, I improved my score to 6.

---

### Official Review · Reviewer_GDLx · 2021-07-16

**Rating:** 6
**Confidence:** 3

**Summary:**

This paper presents an extension of population based bandits (PB2) and the proposed method can handle both continuous variables and categorical variables at the same time. A new algorithm for parallel multi-armed bandits is also presented. The proposed hierarchical approach shows better empirical performance than other approaches. Authors also give theoretical results w.r.t. the cumulative regret bound.

**Limitations And Societal Impact:**

See the above review.

**Main Review:**

Jointly optimizing categorical and continuous parameters  is a very important problem.  I don't have much experience on RL, so the RL parts in this work will not be commented.

advantages of this paper:
1. This paper is well-written.
2. Empirical experiments are detailed and extensive.
3. Sound theoretical results are given w.r.t. the cumulative regret.

limitations of this work:
1. The proposed PB2-Mult treats continuous parameters conditioned on different categorical variables are independent. It is equivalent to model Gaussian processes independently. However, the assumed independence usually doesn't hold in practice.  This explains why PB2-Mult performs best among all competing algorithms in Fig2 and it doesn't perform better than PB2-Mix in Fig3. For the synthetic function, the independence assumption is satisfied while in the RL experiments, possibility such an independence doesn't hold. See [1] for a possible solution, which is also a hierarchical approach and jointly models continuous and categorical variables.
2. Maybe it is common in RL literature, I feel it is strange to use mean to measure performance instead of median in Fig 3 and Table2.  Median is a more common in BO  literature and is more robust.


1.Ma, X., Blaschko, M.: Additive Tree-Structured Covariance Function for Conditional Parameter Spaces in Bayesian Optimization. In: International Conference on Artificial Intelligence and Statistics. pp. 1015–1025. PMLR (2020).



**Time Spent Reviewing:**

4

---

> ### Author Response · Authors · 2021-08-06
> **Thank you for your review!**
>
> We thank you for your generally positive review of our work. We appreciate your expertise lies outside of RL, but we are glad you still found the work to be well-written, with strong empirical and theoretical results. We hope we can clarify your issues and provide you with sufficient confidence to raise to an accept.
>
> Regarding the two limitations mentioned:
>
> **PB2-Mult vs. Mix in different settings**
>
> We agree, this work is the first of its kind to use model based hyperparameter optimization for both continuous and categorical variables *on the fly* in an RL setup. However, the two methods we propose have different properties and it is likely that for some problems one would be significantly more desirable than the other. For example, if we have a large population size but only a small number of categories, then fitting individual GPs may make sense because each GP would have a large dataset. If we have many categories and a small batch/population size, then we likely want to use PB2-Mix.
>
> Given the framework we have here, it is possible to improve both of these methods using the latest ideas from the BO/blackbox optimization community. The paper you linked looks like an interesting future direction and we will think about using it in our next work which will be extending this further. In addition, another recent algorithm [1] extended CoCaBO to higher dimensional input spaces, using trust regions, and this could be directly applied to PB2-style algorithms in the framework we provided.
>
> **Using the mean vs. median**
>
> This makes sense, we used the mean just to follow Raileanu et al (2020), and others on Procgen. However, we view this line of work as a possible bridge between the BO/GP-bandit community and RL, so it is helpful to know that median is used more in that community. With that in mind, see below our Procgen results with the median for each individual task (out of 5 seeds) and then the mean of the medians w.r.t PB2-Rand.
>
> | Environment | PB2-Rand | PB2-Mix  | Improvement |
> |---|---|---|---|
> | BigFish  | 8.2 |  10.8 | +32\% |
> | CaveFlyer | 6.0 |  6.1 | +2\% |
> | CoinRun | 7.7 | 8.2 | +6\% |
> | FruitBot  | 30.5 | 29.1 | -5\% |
> | Jumper  | 5.7 | 6.0  | +5\% |
> | Leaper  | 5.5 | 6.6  | +20\% |
> | StarPilot | 33.1 | 36.6  | +11\% |
> | Mean | 100 | 110$\pm$12  | +10\% |
>
> This again confirms that our method provides an improvement on the recently introduced PB2-Rand. The only environment where it is not the case is FruitBot, which we showed in Fig 4.a) does not seem to exhibit significant dependence between the categorical and continuous variables. Also note in the UCB-DrAC paper that most augmentations perform equally well on FruitBot, so this is consistent with their findings too. We can include this table in the Appendix, or if the Reviewer believes it is important, we can possibly move it to the main body. The only challenge is we do not have the medians for PPO or UCB-DrAC as those were taken from the results table in the paper.
>
> Overall - we understand the reviewer is not an expert on RL, however, we hope that with our clarifications and the reviewer’s expertise in BO, it is possible to increase the score to an accept. If this is not possible yet, then please let us know which areas of concern remain.
>
> [1] Think Global and Act Local: Bayesian Optimisation over High-Dimensional Categorical and Mixed Search Spaces. Wan et al, ICML 2021.

---

### Author Response · Authors · 2021-08-16
**Check in**

Hi all,

Thank you again for your time.

It has been around ten days since we posted responses to your reviews, and we feel we have addressed all of the concerns raised. If this is not the case, please let us know so that we can have the opportunity to discuss further.

Thank you!

Paper 5357 Authors

---

### Decision · Program_Chairs · 2021-09-27

**Decision:**

Accept (Poster)

**Comment:**

There was extensive discussion between the authors and reviewers, and amongst the reviewers privately. Many concerns were raised, and largely addressed. This is particularly the case with the number of runs and statistical significance, which is a concern that has largely been allayed.

The primary remaining concern is simply in the simplicity of the BO (Bayesian optimization) results. An expert in BO assessed the BO theory to be relatively solid, but not particularly exciting in the BO literature. The claims about novelty for addressing both categorial and continuous variables is a bit overstated, considering there are several works that already address this problem in BO (that can be used here). This work extends an existing algorithm, developed for the RL setting, that only works for continuous hyperparameters, so the contribution is placed relative to that. But, the limitation of this previous approach does not imply that these existing BO algorithms could not have already been applied to RL to solve this issue of mixed hyperparameters. This claim on the technical novelty should be clarified, especially given that there appears to be more effective BO algorithms for mixed hyperparameters.

Additionally, the work begins by highlighting the importance of making RL more usable in practice. But, the work focuses on the setting where you can have populations of agents learning in parallel. The primary setting where it is feasible to train many agents in parallel is in simulation. Many real-world RL settings are restricted to a single stream of experience. This contrasts AutoML, which is typically done on a dataset, and so many hyperparameters can be tested. In RL, it is often not feasible to test many hyperparameters, except (a) in experiments to better understand our algorithms (not a deployment setting) and (b) learning in simulators (a useful but relatively restricted setting). The paper should better place the problem setting for which you are developing AutoRL algorithms. This statement may seem to be more generally about AutoRL (where many papers do not do this); but we are here to discuss and improve this paper, and clarity of problem setting is critical.